# Metformin and insulin exacerbate one-carbon metabolism deficits in pregnant growth restricted rats and impacts the placenta, fetal liver and pancreas

Dayna A. Zimmerman[1] , Jessica F. Briffa[2], Dewei Kong[3,4], Sogand Gravina[2], Dara Daygon[5] , Vinod Kumar[1] , Karen M. Moritz[1] , Lillian Y. Lim[3], Adrian K. K. Teo[3,6,7], Shiao-Yng Chan[8] , Mary E. Wlodek[2,8] and James S. M. Cuffe[1]

[1] *School of Biomedical Sciences, The University of Queensland, St Lucia, QLD, Australia*
[2] *Department of Anatomy and Physiology, The University of Melbourne, Parkville, VIC, Australia*
[3] *Stem Cells and Diabetes Laboratory, Institute of Molecular and Cell Biology, Agency for Science, Technology and Research, Singapore, Singapore*
[4] *Dean's Office, Yong Loo Lin School of Medicine, National University of Singapore, Singapore, Singapore*
[5] *Queensland Metabolomics and Proteomics Facility, The University of Queensland, St Lucia, QLD, Australia*
[6] *Department of Biochemistry and Department of Medicine, Yong Loo Lin School of Medicine, National University of Singapore, Singapore, Singapore*
[7] *Precision Medicine Translational Research Programme, Yong Loo Lin School of Medicine, National University of Singapore, Singapore, Singapore*
[8] *Singapore Institute for Clinical Sciences (SICS), Agency for Science, Technology and Research (A*STAR), Singapore, Singapore*

Handling Editors: Kim Barrett & Max Petersen

The peer review history is available in the Supporting Information section of this article (https://doi.org/10.1113/JP289453#support-information-section).

J. S. M. Cuffe and M. E. Wlodek contributed equally to this work.

The Journal of Physiology

**Abstract figure legend** Uteroplacental insufficiency (growth restricted) or sham (Control) surgery was performed on embryonic day (E)18 in Wistar–Kyoto rats. Female F1 offspring were mated and restricted dams received daily metformin, insulin or vehicle from E13. Although restricted dams did not develop metabolic dysfunction during pregnancy, they exhibited reduced one-carbon metabolism. These changes were exacerbated by both metformin and insulin. In F2 fetuses, plasma one-carbon metabolites were unaffected despite changes in expression of genes involved in one-carbon metabolism and DNA methylation in the placenta and liver. Pancreas development was also affected. Antihyperglycaemic medications exacerbated these changes. Created in BioRender: https://BioRender.com/slqi5ht.

**Abstract**  Fetal growth restriction increases the risk of metabolic conditions such as gestational diabetes mellitus (GDM). One-carbon metabolite and nutrient concentrations are dysregulated in GDM and influenced by antihyperglycaemic medications. However, it remains unclear whether disrupted one-carbon metabolism contributes to GDM onset in growth restricted offspring and whether antihyperglycaemic medications exacerbate this dysregulation. We investigated the effects of growth restriction and antihyperglycaemic treatment on one-carbon metabolism in pregnant rats and their fetuses. Uteroplacental insufficiency (Restricted) or sham (Control) surgery was performed on embryonic day (E)18 in Wistar–Kyoto rats. Female F1 offspring were mated and Restricted dams received daily metformin, insulin or vehicle from E13. Although restricted dams did not develop metabolic dysfunction during pregnancy, they exhibited reduced one-carbon metabolism and a lower $S$-adenosylmethionine:$S$-adenosylhomocysteine (SAM:SAH) ratio, indicating reduced methylation capacity. These changes were exacerbated by both metformin and insulin. In F2 fetuses, plasma one-carbon metabolites were unaffected despite changes in expression of genes involved in one-carbon metabolism and DNA methylation in the placenta and fetal liver. F2 fetuses displayed an elevated pancreatic $\beta$-cell:islet ratio. Antihyperglycaemic medications altered the expression of multiple one-carbon metabolising enzymes in the maternal liver, the placenta junctional zone and the fetal liver. Metformin also increased pancreatic $\alpha$-cell area. This study suggests disrupted one-carbon metabolism may underly programmed metabolic dysfunction and highlights the need for monitoring females born small. Both metformin and insulin induced similar physiological changes indicating that one is not safer than the other. Treatment decisions should consider potential impacts on long-term health.

(Received 11 June 2025; accepted after revision 5 August 2025; first published online 24 August 2025)

**Corresponding author** J. S. M. Cuffe: School of Biomedical Sciences, The University of Queensland, St Lucia, QLD 4072, Australia.    Email: j.cuffe1@uq.edu.au

## Key points

- Being born growth restricted predisposes females to develop gestational diabetes (GDM) contributing to intergenerational transmission of disease. GDM is often treated with metformin or insulin. Disruptions to one-carbon metabolism caused by growth restriction, insulin or metformin may explain transmission of disease.
- This study investigated the effects of growth restriction, metformin and insulin on one-carbon metabolism and subsequent fetal outcomes in rats.
- Growth restriction impaired one-carbon metabolism in mothers resulting in a reduced $S$-adenosylmethionine:$S$-adenosylhomocysteine ratio, changes to expression of associated enzymes in the placenta and fetal liver and an increase in the fetal $\beta$-cell: pancreatic islet ratio.
- Metformin and insulin exacerbated deficits in one-carbon metabolism in growth restricted dams and their fetuses, whereas metformin also increased pancreatic $\alpha$-cell area.
- This study demonstrates one-carbon metabolism as a key regulator of programmed metabolic disease in pregnancy and informs about the appropriate treatment of GDM in women born growth restricted.

## Introduction

Intrauterine growth restriction (IUGR) affects up to 7% of pregnancies in high-income countries (Romo et al., 2009) and is most commonly caused by uteroplacental insufficiency (Henriksen & Clausen, 2002). Our laboratory has previously utilised a rat model of IUGR (Gallo et al., 2012; Simmons et al., 2001; Wlodek et al., 2005, 2008) and demonstrated that first generation (F1) growth restricted (Restricted) rats have pancreatic $\beta$-cell, nephron and cardiomyocyte deficits. Although male F1 offspring develop impaired glucose tolerance and insulin secretion in adulthood (Siebel et al., 2008, 2010), female F1 offspring appear protected until pregnancy unmasks a loss of glucose control (Gallo et al., 2012; Mahizir et al., 2020). This suggests that growth restricted females have altered metabolic adaptations to pregnancy, with a phenotype similar to what occurs in patients with gestational diabetes mellitus (GDM). In humans, it is well established that intrauterine conditions can increase the future risk of female offspring developing GDM (Claesson et al., 2007). GDM is defined as the onset of hyperglycaemia first recognised during pregnancy (American Diabetes Association, 2018). It remains unknown how growth restriction programmes an increased risk of GDM in offspring. The negative effects of growth restriction in rats were also transmitted to the next generation (F2 offspring) with F2 males at 6 months of life having a $\beta$-cell deficit and a decreased first-phase insulin response (Cheong et al., 2016; Tran et al., 2013). The processes responsible for this transgenerational transmission require further investigation.

Altered one-carbon metabolism is a key biochemical process that may partly mediate offspring disease in response to prenatal adversity (Padmanabhan & Watson, 2013). Furthermore, perturbed one-carbon metabolism has been linked with an increased risk of developing GDM (Fernandez-Osornio et al., 2022; Williamson et al., 2022). As such, it is possible that programmed deficits in offspring one-carbon metabolism may increase the risk of GDM in females who were born growth restricted.

One-carbon metabolism involves interrelated folate and methionine cycles, which collectively regulate biological processes including methylation, biosynthesis, amino acid homeostasis and redox defences. Optimal one-carbon cycle function relies on plasma concentrations of key micronutrients and metabolites, including folate, vitamin B12, betaine, choline, glycine, methionine and homocysteine. Imbalances in these components have been linked to GDM (Williamson et al., 2022), poor placental development and impaired fetal growth (Nema et al., 2022; Vanhees et al., 2014). Indeed, we have previously demonstrated disrupted one-carbon metabolism in pregnant rats that were exposed to ethanol around conception, which went on to give birth to growth restricted pups (Steane et al., 2022, 2023). It is possible that one-carbon metabolism dysregulation may be prenatally programmed in mothers who were growth restricted and contribute to perturbed glucose control in F1 mothers and impairments in the F2 pups. Furthermore, maternal metabolic perturbations are postulated to affect fetal pancreatic development (Alejandro et al., 2014), potentially via disruptions to one-carbon metabolism within fetal tissues, which may increase the offspring's risk of metabolic disease. Given the fact that we observed pancreatic $\beta$-cell mass reduction at 6 months in F2 males born to growth restricted mothers who had glucose impairment in their pregnancy (Cheong et al., 2016), it is possible that changes to one-carbon metabolism may be linking these outcomes.

An alternative explanation for the clinical relationship between GDM and altered one-carbon metabolism is that pharmacological treatment of GDM with anti-hyperglycaemic medications could impact this cycle. Metformin (1,1-dimethylbiguanidde hydrochloride), an oral biguanide insulin sensitiser, is a well-established antihyperglycaemic agent for type 2 diabetes mellitus (T2DM) and is used as one of the first-line medications for GDM. Some have suggested that metformin induces better outcomes in comparison to insulin (Zhao et al., 2020) but there remains hesitancy around the use of metformin during pregnancy because of the uncertainty around its mechanism of action as well as emerging evidence of perturbed offspring physiology (Nguyen et al., 2018; Tarry-Adkins et al., 2019). In regard to one-carbon metabolism, studies have demonstrated that metformin treatment during pregnancy can alter the balance of one-carbon metabolites, such as vitamin B12

**Dayna Zimmerman** is a PhD candidate in the Placental Endocrinology laboratory led by Dr James Cuffe at the University of Queensland. Her research focuses on investigating how metformin treatment for diabetes in pregnancy influences placental cellular energy regulation and nutrient-sensing pathways. She is interested in understanding how modulation of these pathways may contribute to maternal and fetal outcomes.

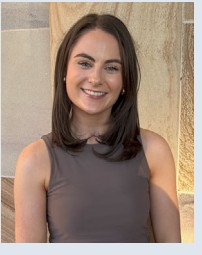

and folic acid, in cord blood, placenta and fetal tissues (Owen et al., 2021). Furthermore, outside of pregnancy, in patients with T2DM given metformin, there are significant decreases in DNA methylation, assumed to be a result of a metformin reducing circulating methionine and *S*-adenosylmethionine (SAM) (García-Calzón et al., 2023). It is often argued that women with GDM are not on metformin long enough to result in a severe deficiency in these metabolites, but this has not been investigated, particularly in rats, which have a much shorter gestation. Furthermore, it is unknown whether the impact of metformin on one-carbon metabolism is any different to the most common alternative: insulin treatment.

The present study aimed to assess one-carbon metabolites and nutrients in pregnant adult rats that were born growth restricted compared to controls. We also aimed to investigate whether one-carbon metabolism is impacted by antihyperglycaemic medications used to treat GDM. To do this, we used a rodent model of growth restricted females with a predisposition for developing glucose intolerance during pregnancy that have been treated with metformin and measured one-carbon metabolites. To determine whether any effects on this pathway were specific to metformin or through changes to glucose control, we also included an insulin treated group. Using this model, we also assessed F2 placental tissue as well as fetal plasma and tissues for changes in one-carbon metabolites, nutrients and pancreatic

deficits. We hypothesise that growth restricted females would have altered plasma one-carbon metabolite and nutrient concentrations during pregnancy. We anticipate that fetuses from these dams will also have lower one-carbon metabolites, which will have consequences on fetal pancreatic development. Furthermore, we hypothesise that short-term antihyperglycaemic treatment with metformin and insulin prenatally would not further affect these disruptions.

## Methods

### Animal procedures

All animal experimentation was approved by The University of Melbourne Animal Ethics Committee (Ethics no. 1714372) following the National Health and Medical Research Councils (NHMRC) Australian code for the care and use of animals for scientific purposes (for details of the model, see Fig. 1). The following experiments comply with the policies and regulations of *The Journal of Physiology*, as described by Grundy (2015). Eight-week-old female Wistar–Kyoto rats were obtained from the Australian Resource Centre (Murdoch, WA, Australia). Rats were housed at 22°C under a 12:12 h light/dark photocycle and *ad libitum* access to food and water. Female rats were mated overnight with normal males and underwent IUGR or sham surgery on E18 as described previously resulting in

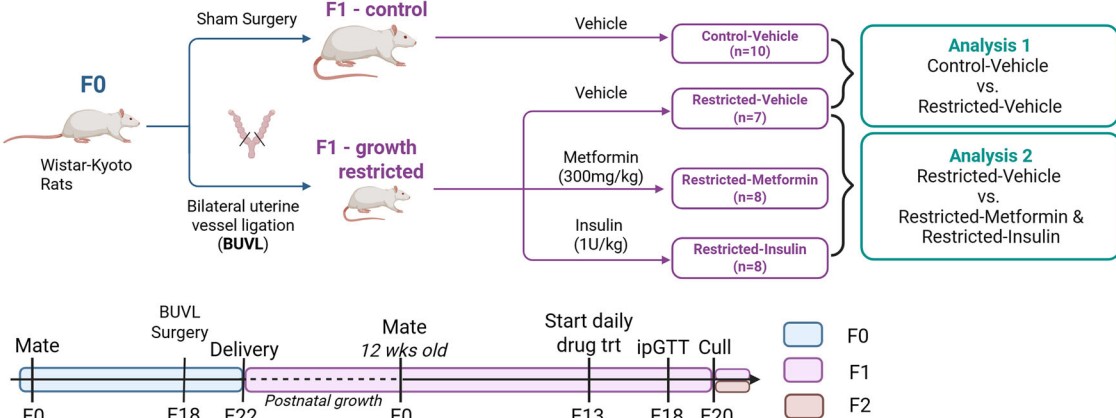

**Figure 1. Flow chart of animal model and experimental groups**
Female Wistar–Kyoto rats (F0) were mated and on Day 18 of gestation (E18; term = 22 days) uteroplacental insufficiency was induced by bilateral uterine vessel ligation (BUVL; offspring termed Restricted). Control animals underwent sham surgery. Dams from both groups were allowed to deliver naturally at term. Postnatal growth was assessed in control and growth restricted offspring, and at 16 weeks of age, F1 Control and Restricted females were mated with normal males. At E13, F1 Restricted dams were randomly allocated to one of three treatment groups: Metformin (300 mg kg$^{-1}$ orally via a syringe), Insulin (glargine s.c. injections, 1 IU kg$^{-1}$) or Vehicle. Metformin vehicle was the metformin compounding agent (orally via syringe) and the insulin vehicle was water (s.c. injections). Animals underwent physiological assessment (IPGTT, metabolic cage, CLAMS) and were culled at E20, with maternal and fetal plasma and tissue collected for analysis. CLAMS, Comprehensive Laboratory Animal Monitoring Systems; E, embryonic; IPGTT, intraperitoneal glucose tolerance test. Created in BioRender: https://BioRender.com/kxkdor9

growth restricted offspring (O'Dowd et al., 2008). In brief, females were anaesthetised with 4% iso-flurane and 650 mL min$^{-1}$ oxygen flow (reduced to 3.2% isoflurane and 250 mL min$^{-1}$ oxygen flow when suturing to aid in the animal's recovery). Animals were monitored throughout surgery to ensure a proper depth of anaesthesia. Uteroplacental insufficiency was then induced by bilateral uterine vessel ligation (offspring termed 'Restricted') or sham (offspring termed 'Control') surgery. Bupivacaine (0.125%) was administered by local infiltration along the edges of the skin and muscle prior to closure for analgesia. Immediately following surgery, the animals were constantly observed until they regained the righting reflex and were then monitored hourly on the day of surgery, then twice daily for 3 days post-surgery. Post-operatively, rats were provided with paracetamol (2 mg mL$^{-1}$) in their drinking water for 48 h post-surgery. Pregnant rats were allowed to deliver naturally at term and birth weights of F1 female offspring were recorded. Offspring were aged to 16 weeks and mated with a control male. At E13, growth restricted F1 females (Restricted) were randomly allocated to receive daily doses of either metformin (300 mg kg$^{-1}$ orally via a syringe) or insulin (glargine daily s.c. injections, 1 IU kg$^{-1}$) or their relevant vehicles. These methods of drug administration mimic the method of exposure in humans (American Diabetes Association, 2003; Du et al., 2022) and doses were clinically relevant (Alfadhli, 2015; Wessels et al., 2014). Given that women without hyper-glycaemia would not be treated with antihyperglycaemic medication, Control rats were given vehicle only and not treated with either metformin or insulin in this model in adherence to the principles of the 3Rs (Grundy, 2015).

### Physical activity, systolic blood pressure and metabolic profiling

On E16, F1 dams were placed into the Comprehensive Lab Animal Monitoring System (CLAMS; Columbus Instruments, Columbus, OH, USA) for 27 h to measure oxygen consumption ($\dot{V}_{O_2}$), carbon dioxide expiration ($\dot{V}_{CO_2}$), respiratory exchange ratio (RER), heat production and spontaneous physical activity, as previously described (Asif et al., 2018; Mahizir et al., 2020). On E18, systolic blood pressure was measured in rats acclimatised to the restraint procedure by tail-cuff plethysmography. On E18, unfasted rats also underwent an intraperitoneal glucose tolerance test (IPGTT). Rats had basal blood glucose measured, followed by an i.p. injection of glucose (1 g kg$^{-1}$; Pharmlab, Lane Cove, NSW, Australia). Tail vein blood samples were collected at 10, 20, 30, 45, 60, 90 and 120 min following the bolus glucose injection (Gallo et al., 2012). Blood glucose was measured at the point of collection with a glucometer with plasma stored at

$-20°C$ for further analysis. Plasma insulin concentrations were measured using a commercial insulin enzyme-linked immunosorbent assay (ELISA) kit (Crystal Chem, Downers Grove, IL, USA). Plasma samples were loaded in duplicate and assays were performed in accordance with the manufacturer's protocols: minimum sensitivity of 0.05 ng mL$^{-1}$ with an intra-assay coefficient of variation of 2.66%. Absorbance was determined using a microplate reader at 450nm and insulin concentrations were calculated using the standard curve of known concentrations. Glucose and insulin area under the curve (AUC) were calculated as the total area under the curve from basal to 120 min for glucose AUC and from basal to 30 min for insulin. Homeostasis model assessment for insulin resistance (HOMA-IR) was calculated using: fasting plasma insulin ($\mu$U mL$^{-1}$) × fasting plasma glucose (mmol:$^{-1}$) ÷ 22.5 (Matthews et al. 1985). On E19, rats were weighed and individually placed in metabolic cages for 24 h measurements of food and water intake and urine production. Rats had been previously acclimatised for 8 h overnight to metabolic cages on E11.

### Post-mortem tissue collection

On E20, F1 dams were anaesthetised with an i.p. injection of ketamine (100 mg kg$^{-1}$ body weight; Parnell Laboratories, Alexandria, NSW, Australia) and Illium Xylazil-20 (30 mg kg$^{-1}$ body weight; Troy Laboratories, Smithfield, NSW, Australia). A terminal cardiac puncture was performed for the rapid collection of maternal blood. The maternal heart, kidneys, liver and pancreas were excised, weighed and corrected to body weight. Fetuses were humanely killed by decapitation, with fetal blood and tails collected. Fetal blood was pooled and plasma stored at $-80°C$. The placenta was separated into the junctional (JZ; endocrine region) and labyrinth zone (LZ; transport region) and the fetal heart, liver, kidney and pancreas were dissected out. All tissues were weighed, snap frozen in liquid nitrogen and stored at $-80°C$. The fetal pancreas was fixed in 10% neutral buffer formalin for histological analysis. Fetal sex was determined using DNA extracted from fetal tails, and SRY expression was measured using quantitative PCR (qPCR) analysis, as previously described (Briffa et al., 2017; Cuffe et al., 2012). Fetal glucose, insulin and glucagon were measured using commercial enzymatic assays/ELISA kits (Crystal Chem). Because limited fetal plasma was available, fewer samples were available for assessment of glucagon compared to glucose and insulin. Plasma glucose, insulin and glucagon levels were also assessed in maternal plasma collected at E20. The glucose kits had an intra-assay coefficient of variation of 6.32% and 10.38% for fetal and maternal assays, respectively. The insulin kit had a minimum sensitivity of 0.05 ng mL$^{-1}$ with an intra- and inter-assay coefficient of variations of

**Table 1. Quantitative PCR primer list.**

| Role | Gene name | Gene symbol | Primer sequence (5'- to 3') |
|---|---|---|---|
| Housekeepers | Beta-actin | *Actb* | F: AAGACCTCTATGCCAACAC |
| | | | R: TGATCTTCATGGTGCTAGG |
| | Hypoxanthine Phosphoribosyltransferase 1 | *Hprt1* | F: ACTGGTAAAACAATGCAGAC |
| | | | R: CCTGAAGTGCTCATTATAGTC |
| | Beta-2-microglobulin | *B2m* | F: ACTGGTCTTTCTACATCCTG |
| | | | R: AGATGATTCAGAGCTCCATAG |
| | TATA box binding protein | *Tbp* | F: CATCATGAGAATAAGAGAGCC |
| | | | R: GGATTGTTCTTCACTCTTGG |
| One-carbon metabolism enzymes | Methylenetetrahydrofolate Reductase | *Mthfr* | F: AAAAAGCTACATCTACCGC |
| | | | R: AAGACCTCAAAGACACTCTC |
| | Methionine Synthase | *Mtr* | F: AGGTCTTCCCAATACCTTTG |
| | | | R: CTTCAAAAACACTATCAGGGG |
| | Methionine Synthase Reductase | *Mtrr* | F: ATTGGTTGCTCCATTTCTTC |
| | | | R: GATAAGTATCTCAGGAGCCAG |
| | Methionine Adenosyltransferase 1A | *Mat1a* | F: GTGTTATTGTCAGGGACTTG |
| | | | R: CTTCTTCCGAAATGACCATAG |
| | Methionine Adenosyltransferase 2A | *Mat2a* | F: GAGATCAGGGTTTGATGTTTG |
| | | | R: TCAGCCAGTTTAGCATTTAG |
| | *S*-adenosylhomocysteine Hydrolase | *Ahcy* | F: GGACAGATTAACCAAGAAAGG |
| | | | R: AATTATACCGTAGGTGACAGG |
| DNA methylation | DNA methyltransferase 1 | *Dnmt1* | F: AGAGACCAGGATAAGAAACG |
| | | | R: TTACTCGTTCAGGTTTCTCC |
| | DNA methyltransferase 3 Alpha | *Dnmt3a* | F: AATAGCCAAGTTCAGCAAAG |
| | | | R: AAACACCCTTTCCATTTCAG |
| | DNA methyltransferase 3 Beta | *Dnmt3b* | F: GATGACAAGGAGTTTGGAATAG |
| | | | R: CAGCGATCTCAGAAAACTTG |
| | Insulin-like Growth Factor 2 | *Igf2* | F: AACCAAACAGGTTTAAGCGC |
| | | | R: CTCCCAGTCGACTGCTTCCAC |

All primers are predesigned KiCqStart SYBR Green primers purchased from Sigma–Aldrich (St Louis, MO, USA).

8.45% and 5.08%, respectively, and the glucagon kit had a minimum sensitivity of 50 pg mL$^{-1}$, and an intra-assay coefficient of variation of 2.86%.

### mRNA expression in maternal liver, placenta and fetal liver

RNA was extracted from 20 mg of tissue using the RNAeasy mini-kit in accordance with the manufacturer's protocol (Qiagen, Melbourne, VIC, Australia). RNA was extracted from maternal liver ($n = 4$–7 per group) as well as the JZ, LZ and fetal liver ($n = 7$–8 per sex per group). RNA yield and purity were confirmed with a Nanodrop spectrophotometer (Thermo Fisher Scientific, Richlands, QLD, Australia). cDNA was synthesised from 500 ng of RNA in a 10-μL reaction using the iScript Reverse Transcription Supermix and C-1000 Thermal Cycler (Bio-Rad, Sydney, NSW, Australia). cDNA was then diluted to 1:10 (5 ng μL$^{-1}$) for use in subsequent analyses. qPCR was performed using the SYBR Green Quanti-Nova PCR Master Mix with predesigned Kiqstart SYBR Green primers (Sigma-Aldrich, Sydney, NSW, Australia). Next, 10-μL reactions containing 10 ng of cDNA were performed in duplicate and gene expression was measured with the QuantStudio 6 Flex PCR system (Thermo Fisher Scientific). The reaction plate was run at 95°C for 2 min then 40 cycles of 95°C for 10 s and 60°C for 20 s. Melt curves were generated following each reaction with all primers producing a single peak. The geometric mean of *Actb* and *Hprt1* was used as the endogenous control for maternal and fetal liver and JZ, with *Actb*, *B2m* and *Tbp* used for the LZ. Analysis of expression was conducted using the $2^{-\Delta\Delta C(t)}$ method with data normalised to the average of the male control group. The present study examined the expression of the following genes: *Mtr*, *Mtrr*, *Mthfr*, *Achy*, *Mat1a* or *Mat2a* (tissue specific), *Dnmt1*, *Dnmt3A*, *Dnmt3B* and *Igf2* (Table 1).

### Liquid chromatography-tandem mass spectrometry (LC-MS/MS)

Concentrations of plasma one-carbon micronutrients and metabolites were measured using separate methods for maternal and fetal plasma. One-carbon micro-

**Table 2. Scheduled multiple reaction monitoring transitions acquired in the LC-MS/MS (positive ionisation).**

| Name | Retention time (min) | Precursor ion (*m/z*) | Product ions (*m/z*) | Q1 Pre bias | Collision energy | Q3 Pre bias | Dwell time (ms) |
|---|---|---|---|---|---|---|---|
| Homocysteine | 2.5 | 135.85 | 90.15 | −13 | −11.5 | −18 | |
| | | | 56.05 | −13 | −17 | −22 | 50 |
| | | | 47.10 | −13 | −28.6 | −19 | |
| Methionine | 3.2 | 150.20 | 104.05 | −12.6 | −11.3 | −21.6 | |
| | | | 56 | −26.8 | −16.4 | −24.2 | 50 |
| | | | 61.1 | −26.8 | −22.5 | −25.5 | |
| *S*-adenosyl Methionine | 6.25 | 399.15 | 250.15 | −15 | −15.7 | −12 | |
| | | | 136.05 | −30 | −27.1 | −14 | 50 |
| | | | 97.1 | −30 | −28.4 | −20 | |
| *S*-adenosyl Homocysteine | 6.5 | 384.95 | 136.05 | −35.8 | −20.3 | −30.6 | |
| | | | 134 | −17.8 | −19.7 | −13.9 | 50 |
| | | | 88.05 | −34.5 | −44.7 | −19 | |
| Folic acid | 6.7 | 442.00 | 295.15 | −29 | −15 | −24 | |
| | | | 176.05 | −29 | −41 | −21 | 50 |
| Cyanocobalamin-M+2H | 6.8 | 678.20 | 147 | −19 | −41.3 | −15.2 | |
| | | | 359.05 | −21.6 | −22.8 | −25.5 | 50 |
| AZT (ISTD) | 7 | 268.00 | 127 | −25.3 | −19.5 | −28.2 | 10 |
| | | | 110.1 | −12.3 | −30.6 | −10.8 | |

MS/MS settings: Nebulizing Gas flow: 3 L min$^{-1}$. Heating gas flow: 10 L min$^{-1}$. Drying gas flow: 10 L min$^{-1}$. Interface temperature: 300°C. Desolvation line temperature: 250°C. Heat block temperature: 400°C.

nutrients and metabolites were measured in maternal plasma ($n = 7$–9 per group) using methods previously described (Neal et al., 2023). Fetal plasma ($n = 5$–7 per group) concentrations of one-carbon micronutrients and metabolites were measured using the API 3200 (AB Sciex, Mulgrave, VIC, Australia) triple quadrupole LC-MS/MS system with positive ionisation. Samples were spiked with the internal standard, methyl-d9-choline and deproteinised with acetonitrile + 2% formic acid. Samples were centrifuged, and supernatants were collected and dried. Samples were resuspended in 50 µL of 75% methanol, with 10 µL analysed on the LC-MS/MS (Table 2). For maternal plasma, homocysteine, B12, methionine, SAM and *S*-adenosylhomocysteine (SAH) were examined with relative metabolite levels normalised to the average of the Control group. For fetal plasma, homocysteine, folic acid, methionine, SAM and SAH were measured against a standard curve.

### Pancreas histology

Fetal pancreatic tissue was processed and embedded in paraffin. Pancreas samples were sectioned and stained in two batches, each containing representative samples from each litter. Blocks contained up to six pancreases from the same litter. Tissues were exhaustively sectioned at 5 µm thickness and the first showing maximal surface area for all six pancreases was used for analysis. Sections were stained with haematoxylin and eosin (H&E) using

a standard protocol or subject to double immuno-fluorescence (IF). This was completed for each fetus from up to eight fetuses per sex, per litter. The average values for male and female fetuses per litter were then included in the analysis. Final sample size includes average male and female data from 7 to 10 litters per group.

For immunofluorescence staining, sections were deparaffinised in xylene and rehydrated in 100%, 95% and 70% ethanol, and demineralised water, then placed in citrate buffer (10 mM citric acid, 0.05% Tween 20, pH6.0) and microwaved for antigen retrieval. Sections were placed in phosphate-buffered saline (PBS) for 10 min at 4°C before fixing with 4% paraformaldehyde (#163-20145; Wako, Osaka, Japan) for 10 min at room temperature. Sections were permeabilised and blocked (0.1% Triton X-100, 5% bovine serum albumin in PBS) for 1 h at room temperature, then sequentially incubated with rabbit anti-insulin (CST3014S; dilution 1:100; Cell Signaling Technology, Danvers, MA, USA) and mouse anti-glucagon (G2654; dilution 1:100; Sigma-Aldrich) antibodies overnight at 4°C followed by corresponding secondary antibodies: anti-rabbit AlexaFlour 488 (A21206; dilution 1:500; Life Technologies, Waltham, MA, USA) and anti-mouse AlexaFlour 594 (A21203; dilution 1:500; Life Technologies) for 1 h at room temperature. Nuclei were stained with 4′,6-diamidino-2-phenylindole (DAPI) (D9542) (dilution 1:5000; Sigma-Aldrich) (3 min at room temperature) and coverslips mounted using the Antifade Mounting Medium (VECTASHIELD, H-1900;

VectorLabs, Newark, CA, USA) and sealed with nail polish. Images were taken using the LSM800 Upright Confocal Microscope (Zeiss, Oberkochen, Germany).

Both H&E and IF images were taken in a blinded fashion in random order. ImageJ (NIH, Bethesda, MD, USA) was used to quantify the proportional area of islets, $\beta$-cells and $\alpha$-cells with thresholding applied to improve precision. For the quantification of the islet proportional area, the islet area and pancreas area were manually outlined. Islet proportional area was calculated as the islet-to-total pancreas area. Insulin, glucagon and DAPI positive areas were manually outlined to calculate $\beta$-cell and $\alpha$-cell proportional areas (insulin or glucagon positive area/DAPI positive area). Islet, $\beta$-cell and $\alpha$-cell proportional area within each batch were median-centred and standardised by $Z$-scoring using the equation: $Z$-score = (actual value – batch median)/batch standard deviation (SD) to facilitate the combination of results from both batches. One slide from each block containing complete pancreases from up to six fetuses was analysed for each litter for both H&E and IF. Outliers were defined as the mean $\pm$ 5SD for batches 1 and 2 separately (two outliers for $\beta$-cells and one outlier for $\alpha$-cells were excluded). $Z$-scores of islets, $\beta$-cell and $\alpha$-cell proportional area were used to calculate the $\beta$-cell-to-islet, $\alpha$-cell-to-islet and $\beta$-cell-to-$\alpha$-cell ratios.

### Statistical analysis

All maternal data ($n = 7$–10 per group) were tested for normality. Two major overarching questions were asked during this study: (1) how did F1 growth restriction impact one-carbon metabolism and (2) how did anti-hyperglycaemic medication given to rats predisposed to develop glucose intolerance impact one-carbon metabolism. As such, statistical analysis was broken down into two groupings. First, Control and Restricted maternal data were analysed using a parametric *t* test. A separate analysis was performed to explore the effect of antihyperglycaemic treatment comparing Restricted rats treated with vehicle, metformin or insulin using one-way analysis of variance (ANOVA). Data that were not normally distributed were analysed using Holm–Sidak's multiple comparison tests. For all fetal and placental data ($n = 7$–10 per sex per group), analyses were performed using two-way ANOVA with Bonferroni's adjustment with group or treatment and sex as main factors. If any significance was found as a result of group/treatment, sex or an interaction between the two, then Sidak's *post hoc* multiple comparisons tests were used. For non-normally distributed fetal or placental data, a non-parametric one-way ANOVA was conducted. If significantly different, *post hoc* analysis was conducted for each sex independently. All data were expressed as the mean $\pm$ SD. $P < 0.05$ was considered statistically significant. Analysis was performed using Prism, version 10.1.2 (GraphPad Software Inc., San Diego, CA, USA).

## Results

### Maternal characteristics

In the current cohort of animals, uteroplacental insufficiency in F0 females did not affect litter size ($P = 0.721$) but did reduce F1 female birth weight ($P = 0.012$) (Table 3). Restricted females remained lighter at postnatal day 35 ($P = 0.003$) but caught up to Controls by the time of mating at 16–20 weeks ($P = 0.618$) (Table 3). There were no differences in body weight ($P = 0.669$), body fat percentage ($P = 0.864$), pregnancy weight gain ($P = 0.677$) or organ weights between Control and Restricted dams at post-mortem (Table 3). Treatment of Restricted animals with daily anti-hyperglycaemic medication from E13 did not impact body weight ($P_{\text{trt}} = 0.203$), body fat percentage ($P_{\text{trt}} = 0.327$), pregnancy weight gain ($P_{\text{trt}} = 0.831$) or organ weights on E20 (Table 4).

### Systolic blood pressure and IPGTT

Systolic blood pressure was not different between Control and Restricted pregnant females at E18 ($P = 0.148$) (Table 3). Pregnant Restricted females were previously shown to have impaired glucose tolerance, demonstrated through reduced insulin AUC and increased glucose AUC in response to IPGTT (Gallo et al., 2012). In this cohort of animals, the second hit of pregnancy did not impair glucose tolerance in the Restricted females, with no difference in glucose AUC ($P = 0.537$) (Table 3), insulin AUC ($P = 0.499$), basal blood glucose ($P = 0.581$) or basal plasma insulin ($P = 0.887$) (Table 3). Plasma glucagon was also not different between Control and Restricted dams on E20 ($P = 0.076$) (Table 3). Treatment of Restricted animals with antihyperglycaemic medication during pregnancy did not impact blood pressure ($P_{\text{trt}} = 0.429$) (Table 4). Given the lack of glucose intolerance in this cohort, it was unsurprising that antihyperglycaemic treatment did not impact basal glucose ($P_{\text{trt}} = 0.584$) or insulin ($P_{\text{trt}} = 0.440$) and did not improve glucose ($P_{\text{trt}} = 0.178$) or insulin ($P_{\text{trt}} = 0.706$) AUC following IPGTT compared to the vehicle treated group (Table 4). Plasma glucagon was not different between vehicle, metformin or insulin treated groups at E20 ($P_{\text{trt}} = 0.492$) (Table 4).

### Food and water intake, renal function, respiratory measurements and physical activity

There was no difference in food ($P = 0.312$), or water ($P = 0.776$) consumption between Control and Restricted dams. Twenty-four-hour urine flow rate ($P = 0.794$)

**Table 3. F1 female growth restricted offspring undergo catch-up growth; once pregnant, body weight/composition, cardiometabolic, renal and respiratory parameters are similar to control animals.**

| | Control (n = 10) | Restricted (n = 7) | P value |
|---|---|---|---|
| F0 litter size | 6.50 ± 1.78 | 6.80 ± 2.18 | 0.721 |
| **F1 body weight (g)** | | | |
| Postnatal day 1 (litter ave) | 4.24 ± 0.33 | 3.78 ± 0.44 | **0.012** |
| Postnatal day 7 | 10.03 ± 1.32 | 9.13 ± 1.74 | 0.157 |
| Postnatal day 14 | 22.36 ± 2.59 | 20.24 ± 2.93 | 0.059 |
| Postnatal day 35 | 79.67 ± 5.49 | 71.75 ± 6.93 | **0.003** |
| Mating | 226.97 ± 20.49 | 222.10 ± 22.94 | 0.618 |
| Post-mortem | 295.98 ± 22.48 | 296.77 ± 24.13 | 0.669 |
| Pregnancy weight gain | 147.30 ± 17.33 | 151.06 ± 18.85 | 0.677 |
| **Body composition (%)** | | | |
| Fat mass | 14.75 ± 2.14 | 15.12 ± 2.67 | 0.864 |
| Lean mass | 76.16 ± 1.14 | 75.55 ± 2.83 | 0.566 |
| Total water | 68.42 ± 2.67 | 69.95 ± 5.61 | 0.487 |
| Bone mass | 9.080 ± 1.40 | 9.322 ± 1.01 | 0.723 |
| **F1 organ weight (% BW)** | | | |
| Heart | 0.31 ± 0.05 | 0.33 ± 0.07 | 0.552 |
| Kidney | 0.52 ± 0.02 | 0.49 ± 0.05 | 0.133 |
| Liver | 4.02 ± 0.18 | 3.88 ± 0.17 | 0.134 |
| Pancreas | 0.43 ± 0.05 | 0.43 ± 0.08 | 0.551 |
| **F1 cardiometabolic measures** | | | |
| Systolic blood pressure (mmHg) | 144.64 ± 2.40 | 150.00 ± 10.24 | 0.148 |
| Basal glucose (mmol $L^{-1}$) | 6.95 ± 0.76 | 7.16 ± 0.73 | 0.581 |
| Basal insulin (ng $mL^{-1}$) | 4.58 ± 1.69 | 4.55 ± 1.48 | 0.887 |
| Glucose AUC (mmol $L^{-1}$ * 120 min) | 1499 ± 188 | 1434 ± 240 | 0.537 |
| HOMA-IR | 35.08 ± 12.78 | 36.05 ± 12.40 | 0.878 |
| Insulin AUC (0–30 min) | 259.17 ± 149.31 | 214.26 ± 76.85 | 0.499 |
| Plasma glucagon (pg $mL^{-1}$) | 163.63 ± 97.79 | 251.41 ± 70.96 | 0.076 |
| **F1 24h metabolic cage measures** | | | |
| Food intake [g $(24 h)^{-1} kg^{-1}$] | 57.37 ± 7.64 | 53.69 ± 5.95 | 0.312 |
| Energy [MJ $(24 h)^{-1} kg^{-1}$] | 1.00 ± 0.14 | 1.32 ± 1.00 | 0.907 |
| Water intake [mL $(24 h)^{-1} kg^{-1}$] | 159.22 ± 48.67 | 175.04 ± 21.60 | 0.776 |
| Urine flow rate [L $(24 h)^{-1} kg^{-1}$] | 0.11 ± 0.05 | 0.11 ± 0.04 | 0.794 |
| Faeces [g $(24 h)^{-1} kg^{-1}$] | 0.02 ± 0.02 | 0.02 ± 0.01 | 0.329 |
| **Respiratory measurements** | | | |
| $\dot{V}_{O_2}$ light cycle (mL $kg^{-1} h^{-1}$) | 1282 ± 114 | 1246 ± 141 | 0.580 |
| $\dot{V}_{O_2}$ dark cycle (mL $kg^{-1} h^{-1}$) | 1556 ± 129 | 1520 ± 198 | 0.653 |
| $\dot{V}_{CO_2}$ light cycle (mL $kg^{-1} h^{-1}$) | 1040 ± 132 | 1015 ± 143 | 0.709 |
| $\dot{V}_{CO_2}$ dark cycle (mL $kg^{-1} h^{-1}$) | 1360 ± 140 | 1330 ± 200 | 0.714 |
| RER light cycle ($\dot{V}_{CO_2}/\dot{V}_{O_2}$) | 0.81 ± 0.03 | 0.81 ± 0.03 | 0.765 |
| RER dark cycle ($\dot{V}_{CO_2}/\dot{V}_{O_2}$) | 0.87 ± 0.02 | 0.87 ± 0.03 | 0.980 |
| Heat light cycle (kcal $h^{-1}$) | 1.69 ± 0.09 | 1.63 ± 0.17 | 0.313^ |
| Heat dark cycle (kcal $h^{-1}$) | 2.08 ± 0.11 | 2.03 ± 0.22 | 0.822^ |
| X-TOT light cycle (counts) | 1718 ± 568 | 1928 ± 538 | 0.454 |
| X-TOT dark cycle (counts) | 4148 ± 1832 | 4086 ± 1093 | 0.938 |
| X-Ambulatory light cycle (counts) | 563 ± 218 | 662 ± 196 | 0.230^ |
| X-Ambulatory dark cycle (counts) | 1382 ± 755 | 1317 ± 415 | 0.840 |
| Z-TOT light cycle (counts) | 904 ± 672 | 742 ± 315 | >0.999^ |
| Z-TOT dark cycle (counts) | 2349 ± 1160 | 2196 ± 1141 | 0.792 |

Data are the mean ± SD. All data were analysed by unpaired *t* test, or, if non-normally distributed by the Mann–Whitney *U* test (denoted by ^). *P* < 0.05 is statistically significant (bold). BW, body weight.

**Table 4. Antihyperglycaemic medication had no effect on maternal body weights/composition, cardiometabolic, renal or respiratory parameters in F1 Restricted dams.**

| | Vehicle (n = 7) | Metformin (n = 8) | Insulin (n = 8) | P value |
|---|---|---|---|---|
| **F1 body weight (g)** | | | | |
| Mating | 222.10 ± 22.94 | 216.89 ± 26.51 | 221.88 ± 14.98 | 0.884ˆ |
| Post-mortem | 296.77 ± 24.13 | 282.96 ± 14.42 | 298.45 ± 15.26 | 0.203 |
| Pregnancy weight gain | 151.06 ± 18.85 | 148.73 ± 24.63 | 144.79 ± 15.94 | 0.831 |
| **Body composition (%)** | | | | |
| Fat mass | 15.12 ± 2.67 | 17.35 ± 3.69 | 15.05 ± 3.37 | 0.327 |
| Lean mass | 75.55 ± 2.83 | 74.15 ± 3.00 | 75.43 ± 2.89 | 0.593 |
| Total water | 69.95 ± 5.61 | 70.05 ± 6.88 | 68.67 ± 3.26 | 0.858 |
| Bone mass | 9.322 ± 1.01 | 8.501 ± 1.16 | 9.521 ± 0.87 | 0.138 |
| **F1 organ weight (% BW)** | | | | |
| Heart | 0.33 ± 0.07 | 0.33 ± 0.10 | 0.29 ± 0.01 | 0.221ˆ |
| Kidney | 0.49 ± 0.05 | 0.51 ± 0.07 | 0.52 ± 0.02 | 0.521 |
| Liver | 3.88 ± 0.17 | 3.94 ± 0.33 | 3.97 ± 0.30 | 0.839 |
| Pancreas | 0.43 ± 0.08 | 0.43 ± 0.07 | 0.43 ± 0.05 | 0.771ˆ |
| **F1 cardiometabolic measures** | | | | |
| Systolic blood pressure (mmHg) | 150.00 ± 10.24 | 149.87 ± 8.62 | 145.27 ± 5.75 | 0.429ˆ |
| Basal glucose (mmol $L^{-1}$) | 7.16 ± 0.73 | 7.03 ± 1.08 | 7.45 ± 0.57 | 0.584 |
| Basal insulin (ng $mL^{-1}$) | 4.55 ± 1.48 | 3.97 ± 1.92 | 4.20 ± 1.83 | 0.440ˆ |
| Glucose AUC (mmol $L^{-1}$ * 120 min) | 1434 ± 240 | 1795 ± 520 | 1543 ± 277 | 0.178 |
| HOMA-IR | 36.05 ± 12.40 | 31.85 ± 20.97 | 34.64 ± 15.58 | 0.537ˆ |
| Insulin AUC (0–30 min) | 214.26 ± 76.85 | 248.95 ± 98.14 | 229.60 ± 39.22 | 0.706ˆ |
| Plasma glucagon (pg $mL^{-1}$) | 251.41 ± 70.96 | 199.01 ± 104.87 | 211.05 ± 61.48 | 0.492 |
| **F1 24h metabolic cage measures** | | | | |
| Food intake [g $(24 h)^{-1} kg^{-1}$] | 53.69 ± 5.95 | 59.39 ± 9.11 | 51.44 ± 9.27 | 0.266ˆ |
| Energy (MJ $[24 h]^{-1} kg^{-1}$) | 1.32 ± 1.00 | 1.16 ± 0.22 | 1.01 ± 0.39 | 0.616ˆ |
| Water intake [mL $(24 h)^{-1} kg^{-1}$] | 175.04 ± 21.60 | 189.55 ± 33.02 | 176.48 ± 39.51 | 0.619ˆ |
| Urine flow rate [L $(24 h)^{-1} kg^{-1}$] | 0.11 ± 0.04 | 0.13 ± 0.03 | 0.12 ± 0.04 | 0.589 |
| Faeces [g $(24 h)^{-1} kg^{-1}$] | 0.02 ± 0.01 | 0.01 ± 0.01 | 0.01 ± 0.01 | 0.260 |
| **Respiratory measurements** | | | | |
| $\dot{V}_{O_2}$ light cycle (mL $kg^{-1} h^{-1}$) | 1246 ± 141 | 1267 ± 125 | 1316 ± 109 | 0.544 |
| $\dot{V}_{O_2}$ dark cycle (mL $kg^{-1} h^{-1}$) | 1520 ± 198 | 1523 ± 149 | 1589 ± 176 | 0.681 |
| $\dot{V}_{CO_2}$ light cycle (mL $kg^{-1} h^{-1}$) | 1015 ± 143 | 1040 ± 83 | 1075 ± 125 | 0.621 |
| $\dot{V}_{CO_2}$ dark cycle (mL $kg^{-1} h^{-1}$) | 1330 ± 200 | 1342 ± 133 | 1385 ± 183 | 0.805 |
| RER light cycle ($\dot{V}_{CO_2}/\dot{V}_{O_2}$) | 0.81 ± 0.03 | 0.82 ± 0.03 | 0.81 ± 0.04 | 0.889 |
| RER dark cycle ($\dot{V}_{CO_2}/\dot{V}_{O_2}$) | 0.87 ± 0.03 | 0.88 ± 0.03 | 0.87 ± 0.03 | 0.785 |
| Heat light cycle (kcal $h^{-1}$) | 1.63 ± 0.17 | 1.63 ± 0.10 | 1.71 ± 0.14 | 0.364ˆ |
| Heat dark cycle (kcal $h^{-1}$) | 2.03 ± 0.22 | 1.98 ± 0.10 | 2.10 ± 0.21 | 0.418ˆ |
| X-TOT light cycle (counts) | 1928 ± 538 | 2106 ± 931 | 1951 ± 774 | 0.887 |
| X-TOT dark cycle (counts) | 4086 ± 1093 | 4511 ± 1243 | 3853 ± 947 | 0.494 |
| X-Ambulatory light cycle (counts) | 662 ± 196 | 651 ± 337 | 623 ± 218 | 0.779ˆ |
| X-Ambulatory dark cycle (counts) | 1317 ± 415 | 1352 ± 429 | 1227 ± 301 | 0.802 |
| Z-TOT light cycle (counts) | 742 ± 315 | 631 ± 339 | 652 ± 379 | 0.725ˆ |
| Z-TOT dark cycle (counts) | 2196 ± 1141 | 1895 ± 548 | 1796 ± 637 | 0.614 |

Data are the mean ± SD. All data were analysed by one-way ANOVA, or, if non-normally distributed, by a Holm–Sidak's multiple comparison tests (denoted by ˆ). $P < 0.05$ is significant (bold). BW, body weight.

and faeces ($P = 0.329$) output were also not different. $\dot{V}_{O_2}$, $\dot{V}_{CO_2}$, RER and heat production rate were not different between Control and Restricted dams ($P > 0.05$). Physical movements were also unaffected ($P > 0.05$) (Table 3). Treatment of Restricted rats with metformin or insulin had no impact on food ($P_{trt} = 0.266$) or water ($P_{trt} = 0.619$) intake, 24 h urine flow rate ($P_{trt} = 0.589$) or faeces ($P_{trt} = 0.260$) output (Table 4). There was also no impact on $\dot{V}_{O_2}$, $\dot{V}_{CO_2}$, RER, heat production or physical movement ($P_{trt} > 0.05$) (Table 4).

**Table 5. F1 growth restriction does not affect F2 fetal, placental and organ weights.**

| | Males | | Females | | Two-Way ANOVA | | |
| --- | --- | --- | --- | --- | --- | --- | --- |
| | Control (n = 10) | Restricted (n = 7) | Control (n = 10) | Restricted (n = 7) | Group | Sex | Interaction |
| Fetal weight (g) | 1.91 ± 0.19 | 1.89 ± 0.17 | 1.80 ± 0.15 | 1.84 ± 0.12 | 0.951 | 0.179 | 0.595 |
| Placental weight (g) | 0.38 ± 0.08 | 0.37 ± 0.03 | 0.36 ± 0.05 | 0.36 ± 0.03 | 0.904 | 0.411 | 0.638 |
| Fetal: placental ratio | 5.19 ± 0.87 | 5.07 ± 0.62 | 5.15 ± 0.66 | 5.15 ± 0.63 | 0.815 | 0.921 | 0.818 |
| Heart (% BW) | 0.47 ± 0.07 | 0.51 ± 0.07 | 0.51 ± 0.06 | 0.49 ± 0.04 | 0.705 | 0.572 | 0.207 |
| Total kidney (% BW) | 0.64 ± 0.04 | 0.63 ± 0.04 | 0.68 ± 0.05 | 0.68 ± 0.07 | 0.905 | **0.021** | 0.763 |
| Liver (% BW) | 6.54 ± 0.67 | 6.51 ± 0.46 | 6.99 ± 0.51 | 6.83 ± 0.60 | 0.648 | 0.068 | 0.743 |

Data are the mean ± SD. All data were analysed by two-way ANOVA with Bonferroni's adjustment with group and sex as main factors. $P < 0.05$ is statistically significant (bold). BW, body weight.

**Table 6. Antihyperglycaemic medication does not affect fetal, placental and organ weights in F2 fetuses from growth restricted dams.**

| | Males | | | Females | | | Two-way ANOVA | | |
| --- | --- | --- | --- | --- | --- | --- | --- | --- | --- |
| | Vehicle (n = 7) | Metformin (n = 8) | Insulin (n = 8) | Vehicle (n = 7) | Metformin (n = 8) | Insulin (n = 8) | Treatment | Sex | Interaction |
| Fetal weight (g) | 1.89 ± 0.17 | 1.89 ± 0.25 | 1.94 ± 0.14 | 1.84 ± 0.12 | 1.80 ± 0.19 | 1.79 ± 0.19 | 0.935 | 0.084 | 0.731 |
| Placental weight (g) | 0.37 ± 0.03 | 0.35 ± 0.05 | 0.35 ± 0.03 | 0.36 ± 0.03 | 0.36 ± 0.02 | 0.32 ± 0.02 | 0.066 | 0.376 | 0.210 |
| Fetal: placental ratio | 5.07 ± 0.62 | 5.49 ± 0.78 | 5.57 ± 0.68 | 5.15 ± 0.63 | 5.03 ± 0.59 | 5.58 ± 0.58 | 0.148 | 0.535 | 0.482 |
| Heart (% BW) | 0.51 ± 0.07 | 0.52 ± 0.09 | 0.50 ± 0.07 | 0.49 ± 0.04 | 0.52 ± 0.03 | 0.50 ± 0.05 | 0.653 | 0.694 | 0.954 |
| Total Kidney (% BW) | 0.63 ± 0.04 | 0.65 ± 0.08 | 0.65 ± 0.04 | 0.68 ± 0.07 | 0.66 ± 0.06 | 0.70 ± 0.05 | 0.593 | **0.032** | 0.542 |
| Liver (% BW) | 6.51 ± 0.46 | 6.72 ± 0.62 | 6.63 ± 0.44 | 6.83 ± 0.60 | 6.34 ± 1.08 | 6.86 ± 0.25 | 0.631 | 0.773 | 0.272 |

Data are mean ± standard deviation. All data were analysed by two-way ANOVA with Bonferroni's adjustment with treatment and sex as main factors. $P < 0.05$ is significant (bold). BW, body weight.

## F2 parameters

F2 body ($P_{group} = 0.951$) and placental ($P_{group} = 0.904$) weights were not altered in fetuses of F1 Restricted mothers (Table 5). Fetal relative heart ($P_{group} = 0.705$), kidney ($P_{group} = 0.905$) and liver ($P_{group} = 0.648$) weights were also unaffected by maternal growth restriction (Table 5). Treatment of Restricted rats with metformin or insulin did not affect F2 fetal ($P_{trt} = 0.935$), placental ($P_{trt} = 0.066$) or organ weights (Table 6).

Overall, neither Restriction nor treatment of Restricted rats with antihyperglycaemic medication impacted maternal physiology or crude measures of fetal growth. Therefore, any changes in one-carbon metabolism are most probably not a consequence of perturbed physiology but rather a consequence of F1 growth restriction or antihyperglycaemic medication treatment.

## F1 one-carbon metabolism

Plasma concentrations of micronutrients and metabolites required for optimal one-carbon cycle function, such as folate, vitamin B12, methionine and homocysteine (Fig. 2*A*), have been shown to be dysregulated in women with GDM (Fernandez-Osornio et al., 2022). Although rats in the current cohort had no detectable changes in glucose control during pregnancy, we still wanted to assess the impact of being born growth restricted on one-carbon metabolism once female offspring became pregnant. Interestingly, we found a 59.27% reduction in relative vitamin B12 levels ($P < 0.001$), a 38.53% reduction in methionine ($P < 0.001$) and a 48.70% reduction in SAM ($P < 0.001$) in the Restricted group compared to the Control group at E20 (Fig. 2*B*). There was also a 66.2% increase in SAH ($P = 0.004$), resulting in a decrease in relative methylation capacity as indicated by the 71.47% reduction in the SAM/SAH ratio ($P < 0.001$) (Fig. 2*B*).

Antihyperglycaemic medication changed the circulating concentrations of all one-carbon metabolites measured at E20 compared to Restricted rats given vehicle. Specifically, relative levels of B12 ($P_{trt} < 0.001$) methionine ($P_{trt} = 0.010$) and SAM ($P_{trt} < 0.001$) were all decreased by antihyperglycaemic medication (Fig. 2*C*). *Post hoc* analysis identified that only insulin decreased B12 ($P < 0.001$) and methionine ($P = 0.006$) levels, whereas both metformin and insulin decreased SAM ($P < 0.05$)

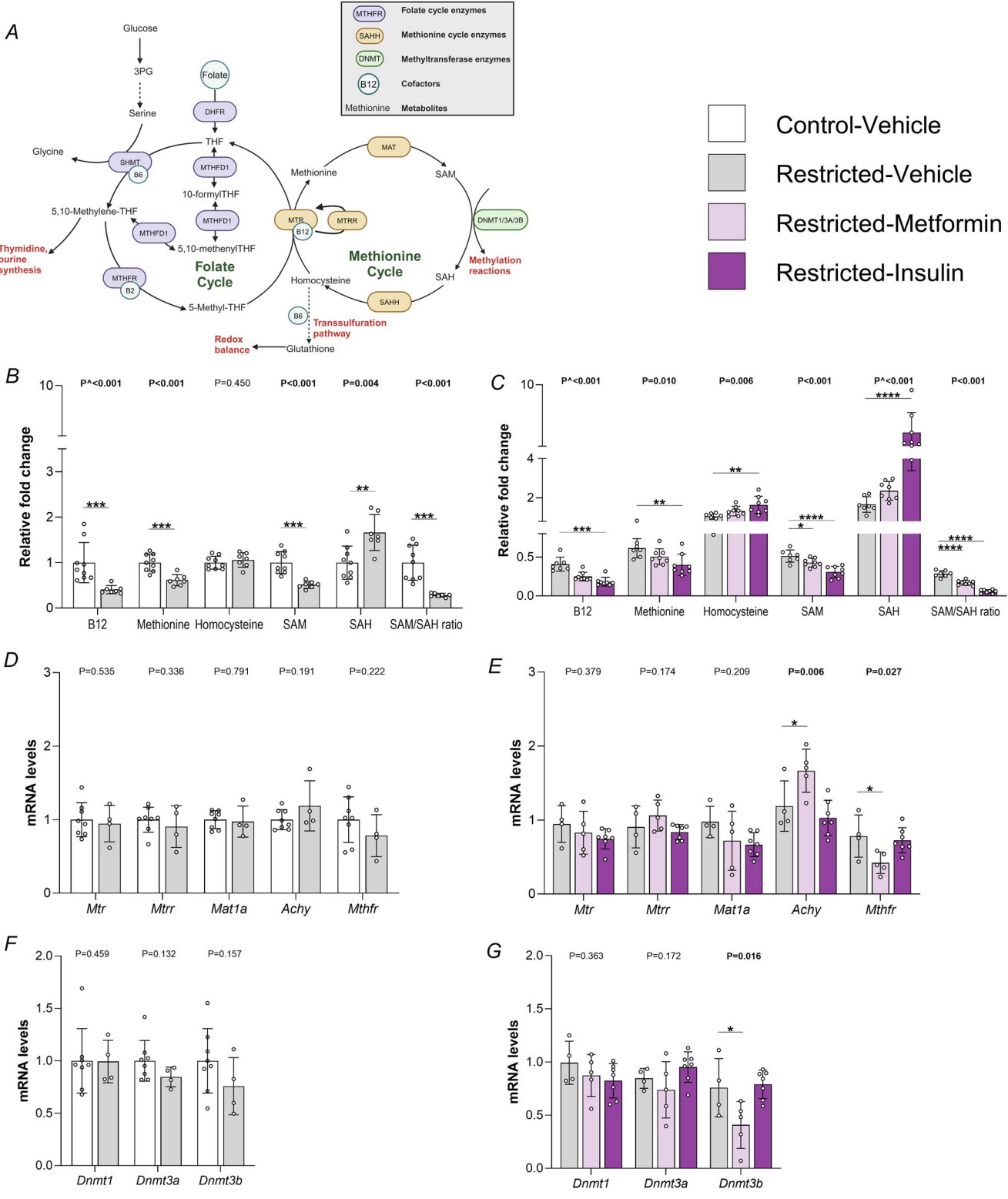

**Figure 2. Maternal one-carbon metabolism**
Effects of maternal growth restriction (Restricted) and antihyperglycaemic medication at embryonic day 20 on one-carbon metabolism (cycle depicted in *A*): *B* and *C*, relative fold change of plasma vitamin B12, methionine, homocysteine, *S*-adenosyl methionine (SAM), *S*-adenosylhomocysteine (SAH) and the SAM/SAH ratio. *D* and *E*, mRNA expression of enzymes involves the methionine-cycle in the maternal liver. *F* and *G*, mRNA expression of DNA methyltransferase genes in the maternal lifer liver. Daily F1 maternal treatment was with either oral dose and s.c. injection of a vehicle, oral dose of metformin (300 mg kg$^{-1}$) or s.c. injection of insulin glargine

(1 IU kg$^{-1}$). Control-Vehicle *vs*. Restricted-Vehicle data were analysed by a parametric *t* test ($n$ = 4–9 per group). Antihyperglycaemic treatment was analysed by one-way ANOVA, or, if non-normally distributed, by a Holm–Sidak's multiple comparison tests (denoted by ˆ) ($n$ = 5–8 per treatment). Data are presented as the mean ± SD. Significant differences between treatment groups indicated by \*$P < 0.05$, \*\*$P < 0.01$, \*\*\*$P < 0.001$ and \*\*\*\*$P < 0.0001$.

(Fig. 2C). Homocysteine ($P_{trt} = 0.006$) and SAH ($P_{trt} < 0.001$) were both increased by treatment, however *post hoc* analysis found that only insulin contributed to this increase. The SAM/SAH ratio was significantly decreased with treatment ($P_{trt} < 0.001$), with a 41.04% and 79.78% reduction in relative methylation capacity in metformin and insulin treated rats, respectively ($P < 0.001$) (Fig. 2C).

The liver plays a major role in regulating systemic concentrations of various one-carbon metabolites (Mato et al., 2013). Given the effect of growth restriction and treatment on circulating concentrations of one-carbon metabolites and nutrients, we next investigated expression of enzymes within the liver which mediate one-carbon metabolism. Restriction had no effect on mRNA expression of methionine synthase (*Mtr*, $P = 0.535$) (Fig. 2D), which is a B12 dependant enzyme that converts homocysteine to methionine, or methionine synthase reductase (*Mtrr*, $P = 0.336$), which reactivates Mtr. Restriction similarly had no effect on methionine adenosyltransferase (*Mat1a*, $P = 0.791$), which converts methionine into SAM, or SAH hydrolase (*Achy*, $P = 0.191$), which converts SAH back into homocysteine. Restriction also had no effect on mRNA expression of methylenetetrahydrofolate reductase (*Mthfr*, $P = 0.222$), a key regulator of the folate and methionine cycle. Although these results suggested that the liver was not the key regulator of systemic changes to one-carbon metabolites, it was still important to determine whether these changes impacted mRNA expression of genes involved in DNA methylation. After all, around 85% of all methylation reactions take place in the liver (Imbard et al., 2021). There was also no difference in the expression of *Dnmt1* ($P = 0.459$), *Dnmt3a* ($P = 0.132$) or *Dnmt3b* ($P = 0.157$) in the Restricted dams compared to Control (Fig. 2F). These results indicate that changes to systemic one-carbon metabolism in Restricted dams may be linked to changes external to hepatic control.

We also assessed mRNA expression of these same genes in livers from animals treated with antihyperglycaemic medication. Maternal exposure to antihyperglycaemic medication altered the expression of *Achy* ($P_{trt} = 0.036$). *Post hoc* analysis identified that metformin increased *Achy* expression (+40.24%, $P = 0.044$) (Fig. 2E), which may partially explain the elevated circulating levels of homocysteine in this group. Antihyperglycaemic medication also altered the mRNA expression of *Mthfr* ($P_{trt} = 0.006$) with *post hoc* analysis demonstrating that metformin decreased *Mthfr* (−46.24%, $P = 0.029$) (Fig. 2E), which may have limited overall flux through

the methionine cycle. Neither metformin nor insulin affected *Mtr* ($P_{trt} = 0.379$), *Mtrr* ($P_{trt} = 0.174$) or *Mat1a* ($P_{trt} = 0.209$) expression (Fig. 2E). Finally, out of the three DNMTs evaluated, only *Dnmt3b* was altered by antihyperglycaemic medication ($P_{trt} = 0.016$). *Post hoc* analysis identified that only metformin decreased *Dnmt3b* expression, by 46.07% ($P = 0.040$) (Fig. 2G).

### Effect of F1 growth restriction on F2 one-carbon metabolism

Despite there being significant differences in most one-carbon metabolites in the F1 dam, there was no effect of Restriction on F2 fetal circulating one-carbon metabolites ($P > 0.05$) (Fig. 3A). The mRNA expression of genes responsible for regulating one-carbon metabolism were also not affected by maternal restriction in the F2 fetal livers ($P_{group} > 0.05$) (Fig. 3B, left). However, inter-actions between group and sex, with *post hoc* analyses, demonstrated that *Mtrr* ($P_{int} = 0.006$, $P = 0.010$) and *Mthfr* ($P_{int} = 0.011$, $P = 0.006$) were elevated in F2 female, but not male, fetuses from Restricted dams by 50.16% and 96.05%, respectively (Fig. 3B). There was no effect of growth restriction ($P_{group} > 0.05$) on any DNMT gene expression in the liver (Fig. 3B, right). A group by sex interaction followed by *post hoc* analysis, however, demonstrated a 48.06% increase in *Dnmt3a* gene expression due to Restriction in the livers of female, but not male, fetuses ($P_{int} < 0.001$, $P = 0.001$) (Fig. 3B).

In addition to the liver regulating concentrations of one-carbon metabolites in maternal circulation, the placenta probably also plays a major role. One-carbon metabolism is crucial in the placenta for providing methyl groups used in various reactions required for placental and fetal development. It may also influence one-carbon metabolites in fetal circulation. Perturbations in one-carbon transfer can have profound effects on cell proliferation, growth and function (Kalhan, 2016); therefore, it was important to investigate the placental expression of these same genes in this study. In the placental JZ, *Mtrr* (−16.60%, $P_{group} = 0.040$) and *Mat2a* (−17.89%, $P_{group} = 0.040$) were decreased in Restricted dams (Fig. 3C, left). There was no sex or group effect on other genes involved in one-carbon metabolism. Given that there was a −49.01% reduction in SAM concentration in the maternal circulation, we also investigated how this impacted placental expression of DNMTs and the methylation responsive gene, *Igf2* (Koukoura et al., 2012). *Dnmt3b* was decreased by maternal growth restriction

## Fetal plasma

*A*

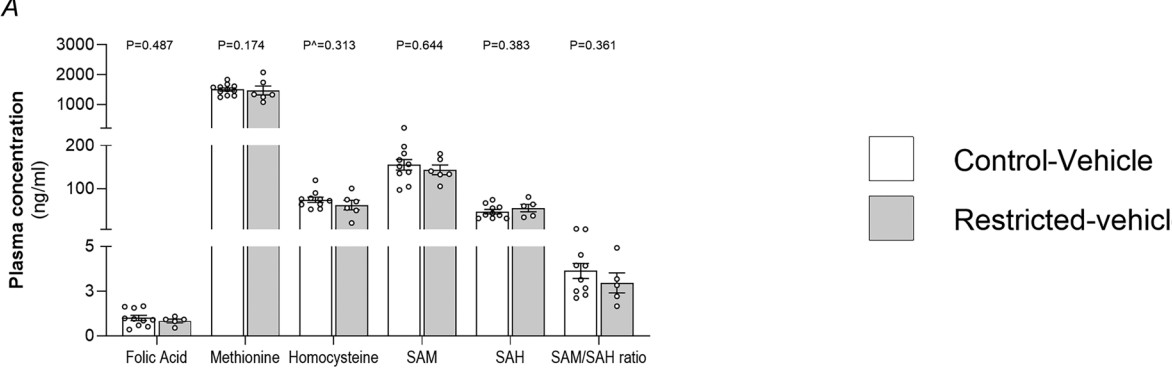

*B* **Fetal Liver**

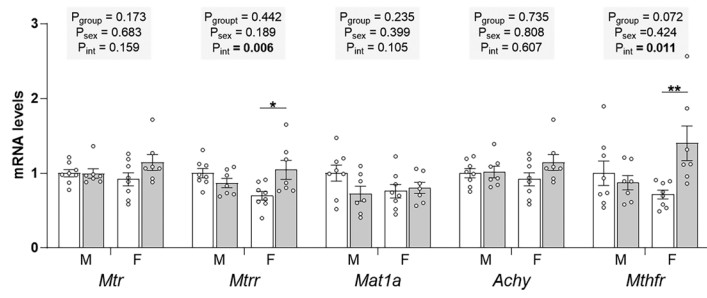

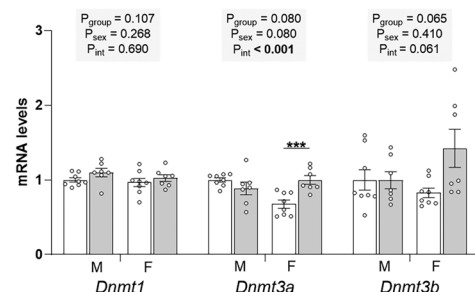

*C* **Junctional zone**

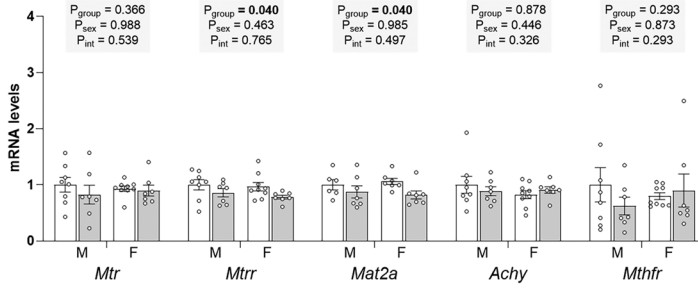

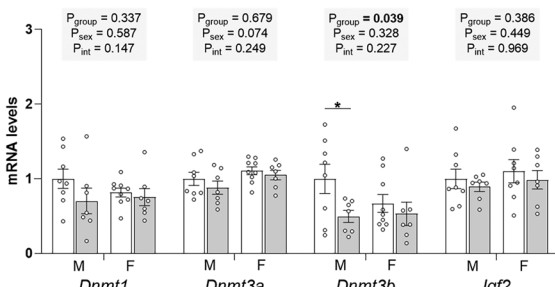

*D* **Labyrinth zone**

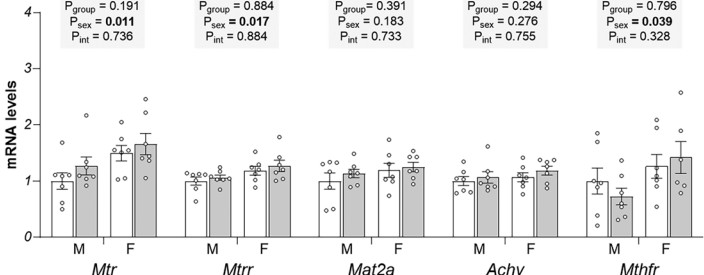

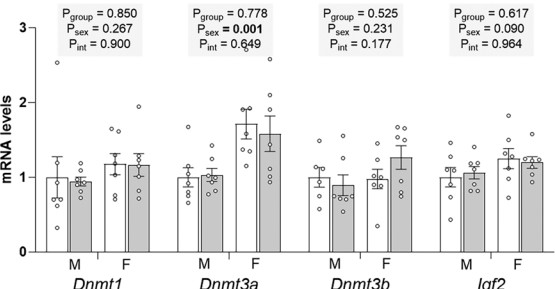

**Figure 3. The effect of maternal growth restriction on fetal one-carbon metabolism**
*A*, plasma concentration of folic acid, methionine, homocysteine, S-adenosyl methionine (SAM),
*S*-adenosylhomocysteine (SAH) and the SAM/SAH ratio; gene expression of methionine-cycle enzymes,

DNA methyltransferases and imprinted genes in the (*B*) fetal liver, (*C*) placental junctional zone and (*D*) placental labyrinth zone at embryonic day 20. Plasma metabolite data were analysed using a parametric *t* test, or, if non-normally distributed, by the Mann–Whitney *U* test (denoted by ˆ) (*n* = 5–10 per group). All other data were analysed by two-way ANOVA with group and sex as major factors. If a significant effect of sex or group or an interaction ($P_{int}$) was detected, then a Sidak's *post hoc* analysis was performed (*n* = 7–8 per sex per group). Data are presented as the mean ± SD. Significant differences within the same sex, between groups, indicated by *$P < 0.05$, **($P < 0.01$ and ***($P < 0.001$.

($P_{group}$ = 0.039), with *post hoc* analysis identifying that this was primarily driven by reduced expression in males (−50.35%) (Fig. 3*C*, right). mRNA expression of other *Dnmt* genes and *Igf2* were unaffected by Restriction ($P_{group}$ > 0.05) or sex ($P_{sex}$ > 0.05) (Fig. 3*C*).

There was no effect of maternal growth restriction on the placental mRNA levels of genes involved in one-carbon metabolism, *Dnmt* genes and *Igf2* in the LZ ($P_{sex}$ > 0.05) (Fig. 3*D*). mRNA expression of *Mtr* ($P_{sex}$ = 0.011), *Mtrr* ($P_{sex}$ = 0.017) and *Mthfr* ($P_{sex}$ = 0.039) was elevated in females compared to males (Fig. 3*D*). *Dnmt3a* expression was also elevated in the LZ of females (+62.36%; $P_{sex}$ = 0.001).

### Effect of maternal metformin and insulin treatment on one-carbon metabolism in the F2 fetus

Despite the significant decrease in methylation capacity found in F1 Restricted dams treated with metformin and insulin, there was no effect on fetal plasma folic acid ($P_{trt}$ = 0.721), methionine ($P_{trt}$ = 0.327), homocysteine ($P_{trt}$ = 0.923), SAM ($P_{trt}$ = 0.871), SAH ($P_{trt}$ = 0.702) or the SAM/SAH ratio ($P_{trt}$ = 0.670) (Fig. 4*A*). However, maternal exposure to antihyperglycaemic medication caused significant changes in gene expression of several one-carbon metabolism enzymes in F2 fetal livers (Fig. 4*B*, left). *Mtr* was decreased by treatment ($P_{trt}$ = 0.026) with *post hoc* analysis showing that insulin reduced *Mtr* expression in females only (−19.77%; *P* = 0.050). The expression of *Mtrr* was not impacted ($P_{trt}$ = 0.386). Although *Mat1a* was altered by treatment ($P_{trt}$ = 0.029) *post hoc* analysis was not able to indicate that this occurred in any specific group. Maternal treatment reduced *Achy* expression ($P_{trt}$ = 0.036) with *post hoc* analysis indicating that insulin reduced its expression in females only (−32.37%). *Achy* mRNA levels were also lower in females compared to males ($P_{sex}$ = 0.015). There was a treatment effect ($P_{trt}$ = 0.006) and a treatment by sex interaction ($P_{int}$ = 0.034) in *Mthfr* expression with *post hoc* analysis showing that metformin increased *Mthfr* in male fetuses (+61.36%; *P* = 0.024), but insulin reduced *Mthfr* expression in female fetuses (−34.60%; *P* = 0.042). Expression of *Dnmt1* was reduced by anti-hyperglycaemic medication ($P_{trt}$ < 0.001) with *post hoc* analysis demonstrating that *Dnmt1* expression was decreased by insulin in both sexes (−19.26%). *Dnmt3a*

was unaffected by treatment ($P_{trt}$ = 0.392), whereas *Dnmt3b* was impacted by antihyperglycaemic medication ($P_{trt}$ = 0.028), although *post hoc* analysis could not determine the overall direction of change (Fig. 4*B*, right).

Maternal exposure to antihyperglycaemic medication decreased the mRNA the expression of multiple one-carbon metabolism enzymes within the placental JZ (Fig. 4*C*, left). Treatment decreased *Mtr* ($P_{trt}$ < 0.001) with *post hoc* analysis demonstrating a ∼50% reduction in expression in both males (*P* = 0.015) and females (*P* = 0.004) (Fig. 4*C*). Treatment decreased *Mat2a* ($P_{trt}$ < 0.001) with *post hoc* analysis demonstrating that metformin decreased *Mat2a* expression in males (−38.63%; *P* = 0.003) and females (−35.16%; *P* = 0.011) (Fig. 4*C*). Treatment also decreased *Mthfr* ($P_{trt}$ = 0.012) with *post hoc* analysis finding that this was driven by both metformin (= 57.44%; *P* = 0.040) and insulin (−63.52%; *P* = 0.022) in females (Fig. 4*C*).

Antihyperglycaemic treatment decreased the expression of *Dnmt1* ($P_{trt}$ = 0.009), *Dnmt3a* ($P_{trt}$ = 0.001) and *Dnmt3b* ($P_{trt}$ < 0.001) in the placental JZ compared to vehicle (Fig. 4*C*, right). *Post hoc* analysis of *Dnmt1* found that this decrease was driven by metformin in female placentas (−50.07%; *P* = 0.029). Meanwhile, the decrease in *Dnmt3a* was driven by both metformin (−27.92%; *P* = 0.009) and insulin (−28.70%; *P* = 0.007) in female placenta. *Dnmt3b* expression in the JZ was decreased by ∼60% by metformin in both males (*P* = 0.020) and females (*P* = 0.015) and by insulin in both males (*P* = 0.017) and females (*P* = 0.020). *Igf2* was significantly decreased in response to treatment ($P_{trt}$ < 0.001) (Fig. 4*C*). *Post hoc* analysis demonstrated that this was driven by a 54%–70% reduction in metformin exposed male (*P* < 0.001) and female (*P* < 0.001) placentas and insulin exposed male (*P* = 0.008) and female (*P* < 0.001) placentas.

Antihyperglycaemic medication treatment did not impact mRNA expression of one-carbon metabolising enzymes or *Dnmt* expression in the LZ ($P_{trt}$ > 0.05). *Mtr* ($P_{sex}$ < 0.001), *Mtrr* ($P_{sex}$ = 0.022), *Mat2a* ($P_{sex}$ = 0.017), *Mthfr* ($P_{sex}$ < 0.001), *Dnmt3a* ($P_{sex}$ < 0.001) and *Dnmt3b* ($P_{sex}$ < 0.001) were all more highly expressed in the LZ of females compared to males. *Igf2 m*RNA expression was increased by treatment ($P_{trt}$ = 0.031), although *post hoc* analysis could not detect where this change had occurred (Fig. 4*D*).

## Fetal plasma

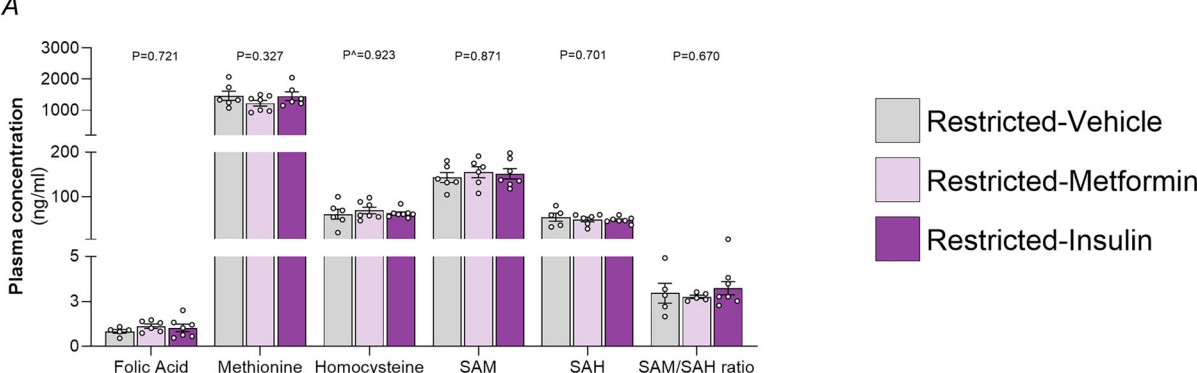

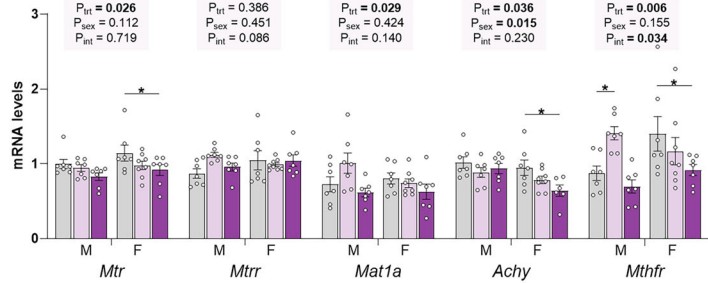
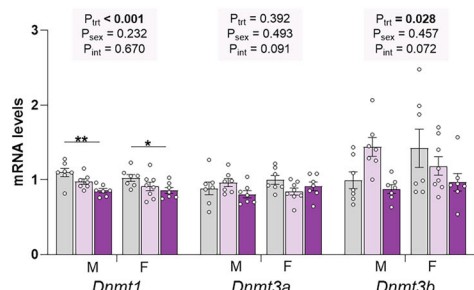

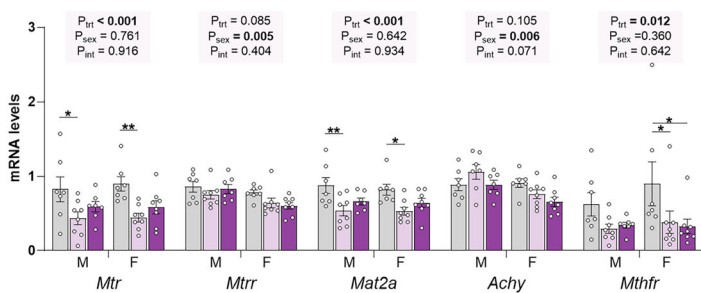
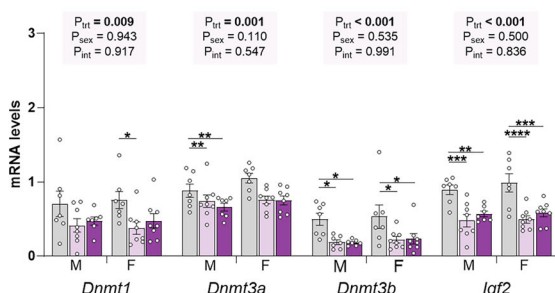

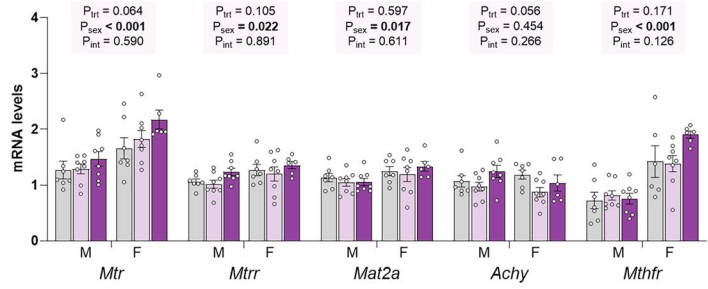
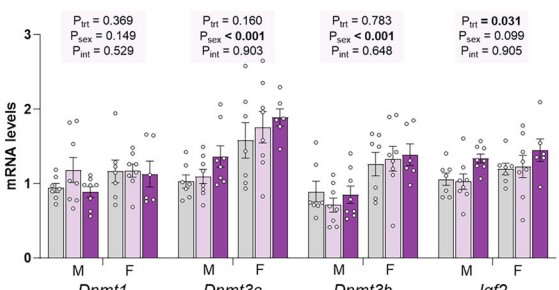

**Figure 4. The effect of maternal antihyperglycaemic medication on fetal one-carbon metabolism**
*A*, plasma concentration of folic acid, methionine, homocysteine, *S*-adenosyl methionine (SAM), *S*-adenosylhomocysteine (SAH) and the SAM/SAH ratio; gene expression of methionine-cycle enzymes, DNA methyltransferases and imprinted genes in the (*B*) fetal liver, (*C*) placental junctional zone and (*D*) placental labyrinth zone at embryonic day 20. Daily F1 maternal treatment was with either oral dose and s.c. injection

of a vehicle, oral dose of metformin (300 mg kg$^{-1}$) or s.c. injection of insulin glargine (1 IU kg$^{-1}$). Plasma metabolite data were analysed using a one-way ANOVA, or if non-normally distributed, by a Holm–Sidak's multiple comparison tests (denoted by ^) ($n$ = 5–7 per group). All other Data were analysed by two-way ANOVA with treatment and sex as major factors. If a significant effect of sex or treatment ($P_{trt}$) or an interaction ($P_{int}$) was detected, then a Sidak's *post hoc* analysis was performed ($n$ = 7–8 per sex per treatment). Data are presented as the mean ± SD. Significant differences within the same sex, between treatments, indicated by *$P$ < 0.05, **($P$ < 0.01, ***($P$ < 0.001 and ****$P$ < 0.0001.

### Fetal pancreas and glucose control

Plasma glucose ($P$ = 0.427), insulin ($P$ = 0.930) and glucagon ($P$ = 0.502) concentrations in the F2 fetal plasma were not different between Control and Restricted dams (Fig. 5A–C). There was no effect of Restriction on F2 fetal $\beta$-cell ($P_{group}$ = 0.853) (Fig. 5D), $\alpha$-cell ($P_{group}$ = 0.158) (Fig. 4E) or islet ($P_{group}$ = 0.235) (Fig. 5G) proportional area. Similarly, the $\beta$-cell/$\alpha$-cell ratio was not affected ($P_{group}$ = 0.395) (Fig. 5F). Because the denominator used encompassed the entire pancreas and considering that islets only comprise a small proportion of the entire pancreas, it is possible that small potential differences in the $\beta$-cell and $\alpha$-cell area could be obscured by this method. Hence, the $\beta$-cell/islet ratio and $\alpha$-cell/islet ratio were also evaluated. This revealed a significant increase in the $\beta$-cell/islet ratio in the Restricted group compared to Control ($P_{group}$ = 0.018) (Fig. 5H), but not that of the $\alpha$-cell/islet ratio ($P_{group}$ = 0.155) (Fig. 5I), suggesting a developmental response of the insulin-expressing $\beta$-cells to counter a potential reduction in $\beta$-cell function (Bihoreau et al., 1986).

Antihyperglycaemic treatment also did not affect fetal plasma glucose ($P$ = 0.621), insulin ($P$ = 0.364) or glucagon ($P$ = 0.318) (Fig. 6A–C). There was no effect of antihyperglycaemic medication on the proportional $\beta$-cell ($P_{trt}$ = 0.169) or islets ($P_{trt}$ = 0.424) cell areas, although female fetuses had a greater $\beta$-cell proportional area (+229.69%; $P_{sex}$ = 0.049) than males. The $\alpha$-cell proportional area was, however, increased by treatment ($P_{trt}$ = 0.019) with *post hoc* analysis demonstrating that this was driven by metformin treatment in female fetuses ($P$ = 0.012) (Fig. 6E). There was no effect of antihyperglycaemic medication on the $\beta$-cell/$\alpha$-cell ($P_{trt}$ = 0.164), $\beta$-cell/islet ratio ($P_{trt}$ = 0.128) or $\alpha$-cell/islet ($P_{trt}$ = 0.172) ratios.

### Discussion

Our previous studies have demonstrated that female growth restricted rats are at an increased risk of developing diabetes during pregnancy (Gallo et al., 2012), which programs F2 metabolic dysfunction (Cheong et al., 2016; Tran et al., 2013). The underlying biological process responsible for this remains unknown. One-carbon metabolism is one potential pathway that is dysregulated in women with GDM (Fernandez-Osornio et al.,

2022) and thus was investigated in the present study. Additionally, the antidiabetic medication metformin, known to disturb one-carbon metabolism outside of pregnancy (Aroda et al., 2016; Kim et al., 2019), was examined for its potential impact on one-carbon metabolism in pregnant growth restricted rats. To determine whether any effects on this pathway were specific to metformin or through changes to glucose control, we also included an insulin treated group.

Our findings confirmed our hypothesis that being born growth restricted reduces concentrations of most one-carbon metabolites during pregnancy. These reductions occurred despite no detectable changes in glucose tolerance in the F1 pregnant dams. By contrast to our hypothesis, antihyperglycaemic medication further exacerbated reductions in one-carbon metabolites even with a relatively short exposure of 7 days. We also demonstrated that Restriction reduced placental expression of key genes involved in the formation of SAM, which suggests that changes in placental regulation of one-carbon metabolism may have contributed to the reduced concentrations of one-carbon metabolites in maternal circulation. Meanwhile, antihyperglycaemic medications impacted the expression of one-carbon metabolism genes in both the maternal liver and the placenta, and it is possible that both tissues contributed to the reduction in circulating metabolites in these dams. Despite these significant effects of Restriction and anti-hyperglycaemic medication on maternal one-carbon metabolites, neither factor influenced fetal circulating metabolite concentrations when measured at E20. We did, however, demonstrate that both Restriction and antihyperglycaemic medications caused changes to the expression of genes involved in DNA methylation in the placenta and fetal liver. We also demonstrated that Restriction and antihyperglycaemic medications impacted fetal pancreas development. This may explain transmission of diabetic-like symptoms to adult F2 offspring and highlights that metformin and insulin treatment may cause long-term health outcomes in offspring.

### Growth restriction and one-carbon metabolism

Clinical studies have demonstrated that women with GDM have altered concentrations of circulating one-carbon metabolites and cofactors (Barzilay et al.,

2018; Lai et al., 2018; Sukumar et al., 2016). Because these metabolites are vital for processes such as DNA methylation, imbalances have been linked with poor placental development and adverse fetal outcomes (Padmanabhan & Watson, 2013; Vanhees et al., 2014). In the present study, growth restricted females had undergone catch-up growth and were of similar maternal body composition and metabolic health to controls.

Despite this, these dams had a significant decrease in multiple one-carbon metabolites during pregnancy, similar to what is reported clinically in women with GDM (Fernandez-Osornio et al., 2022; Sukumar et al., 2016). Specifically, vitamin B12 and methionine were decreased and SAH was increased. Furthermore, despite no changes in liver gene expression of one-carbon enzymes in Restricted dams, there was a 49.01% decrease

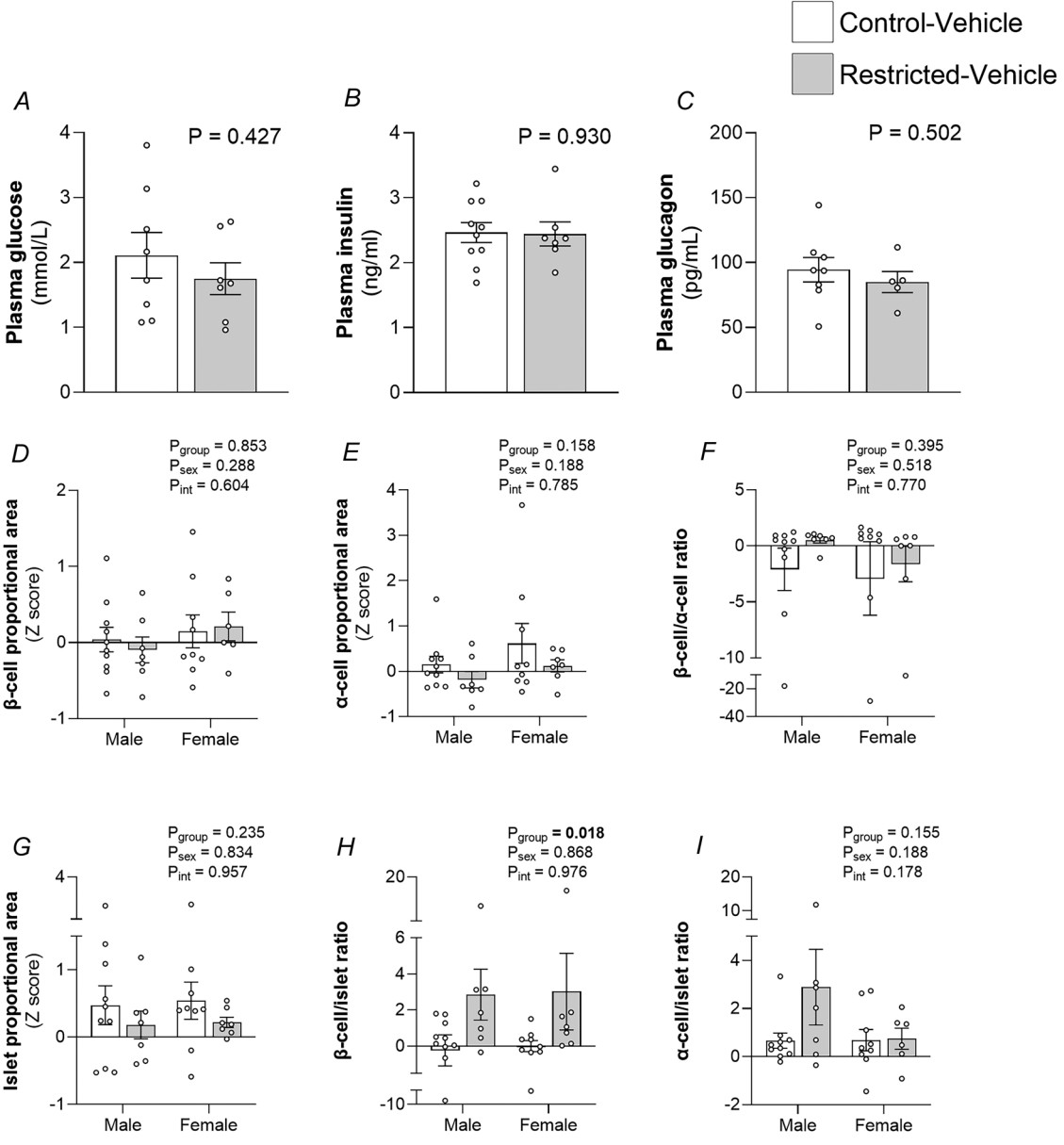

**Figure 5. The effect of maternal growth restriction on fetal glucose control and pancreatic development**
Pooled plasma concentrations of (*A*) glucose (mmol L$^{-1}$), (*B*) insulin (ng mL$^{-1}$), (*C*) glucagon (pg mL$^{-1}$) on embryonic day 20 (E20). Immunofluorescence staining was used to quantify fetal (*D*), $\beta$-cell area, (*E*) $\alpha$-cell area and (*F*) $\beta$-cell/$\alpha$-cell ratio. Haematoxylin-eosin staining was used to determine (*G*) $\beta$-cell area. (*H*) $\beta$-cell /islet ratio and (*I*) $\alpha$-cell /islet ratio were also evaluated. Plasma data were analysed using Student's *t* test (*n* = 5–10 per group). Pancreatic data were analysed by two-way ANOVA with group and sex as major factor. If a significant effect of sex or group or an interaction ($P_{int}$) was detected, then a Sidak's *post hoc* analysis was performed (*n* = 7–8 per sex per group). Data are presented as the mean $\pm$ SD.

in circulating SAM. This led to a significant reduction in overall methylation capacity given the low SAM:SAH ratio. This suggests that, within the maternal system, there is a greater chance for DNA hypomethylation (James et al., 2002). Interestingly, a previous study exploring the effects of growth restriction on one-carbon metabolites in rat neonates found that liver concentrations of homocysteine and SAH were increased: indicating reduced remethylation capacity. They also demonstrated that growth restriction reduced methionine concentrations

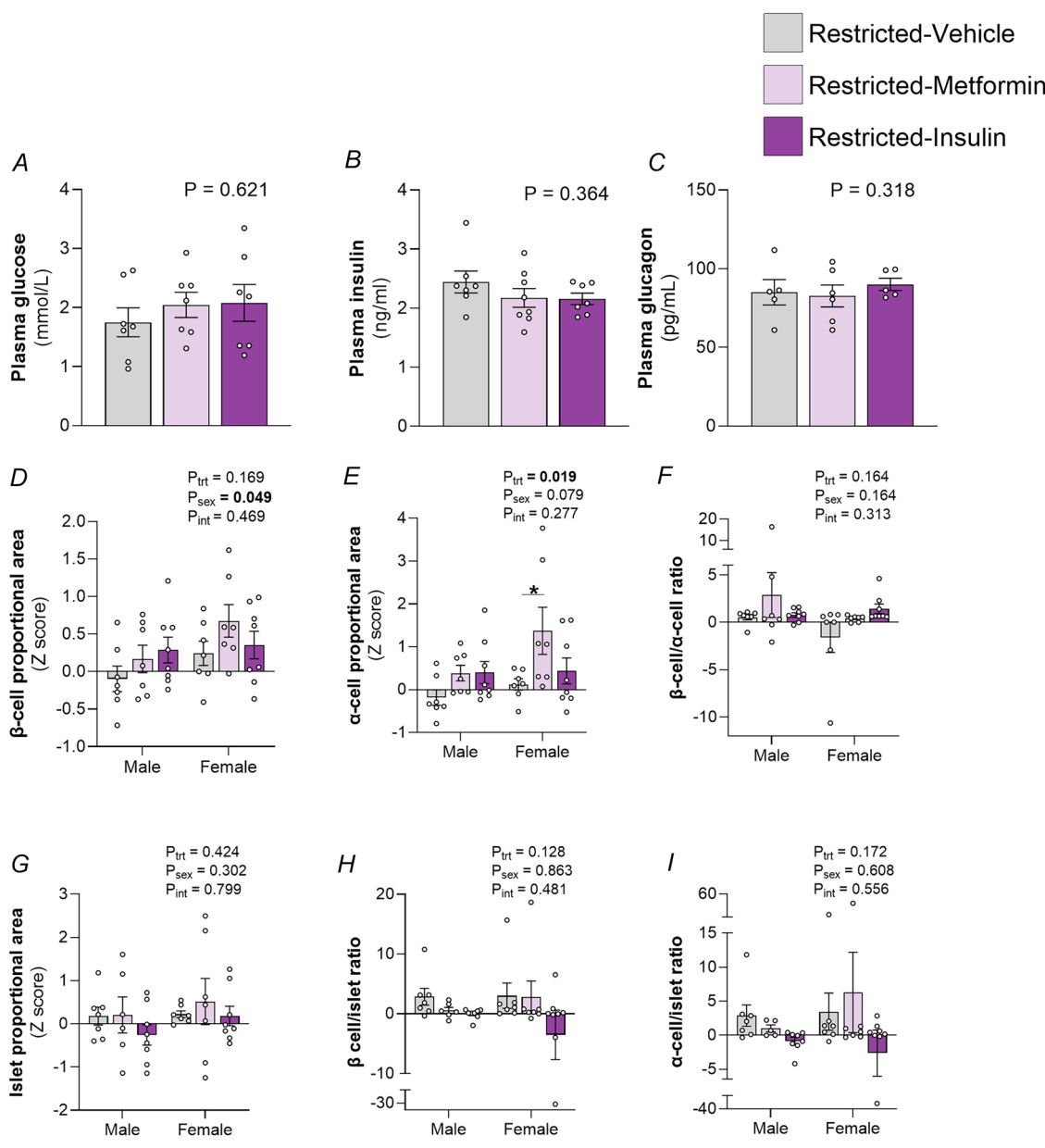

**Figure 6. The effect of maternal antihyperglycaemic medication on fetal glucose control and pancreatic development**
Pooled plasma concentrations of (*A*) glucose (mmol L$^{-1}$), (*B*) insulin (ng mL$^{-1}$), (*C*) glucagon (pg mL$^{-1}$) on embryonic day 20 (E20). Immunofluorescence staining was used to quantify fetal (*D*), $\beta$-cell, (*E*) $\alpha$-cell area and (*F*) $\beta$-cell /$\alpha$-cell ratio. aematoxylin-eosin staining was used to determine (*G*) islet area. (*H*) $\beta$-cell /islet ratio and (*I*) $\alpha$-cell /islet ratio were also evaluated. Daily F1 maternal treatment was with either oral dose and s.c. injection of a vehicle, oral dose of metformin (300 mg/kg) or s.c. injection of insulin glargine (1 IU kg$^{-1}$). Plasma data were analysed using a one-way ANOVA (*n* = 5–10 per group). Pancreatic data were analysed by two-way ANOVA with treatment and sex as major factors. If a significant effect of sex or treatment ($P_{trt}$) or an interaction ($P_{int}$) was detected, then a Sidak's *post hoc* analysis was performed (*n* = 7–8 per sex per treatment). Data are presented as the mean ± SD. Significant differences within the same sex, between treatments, indicated by *$P$ < 0.05.

alongside reduced *Mat* and *Dnmt* expression. Such changes may be expected to result in decreased genomic DNA methylation (MacLennan et al., 2004). The findings from the present study therefore suggest that the decrease in one-carbon metabolism previously reported in growth restricted pups persists into adulthood and remains throughout pregnancy.

Despite these changes in maternal circulating one-carbon metabolites in Restricted females, one-carbon components in F2 fetal plasma were unchanged, suggesting that any deficits in F2 development are not directly regulated by changes in fetal circulating one-carbon components. Changes in maternal one-carbon metabolism may instead impact maternal processes which mediate disease transmission to F2 offspring. Indeed, Jankovic-Karasoulos et al. (2021) have demonstrated that changes to various one-carbon metabolites and nutrients in mothers is linked to multiple pregnancy complications that are known to impair fetal development and consequently lead to offspring disease.

It is interesting to note that despite no changes in fetal one-carbon metabolites, *Mtrr* and *Mthfr* were increased in the liver of Restricted F2 female fetuses compared to Control. These changes may persist beyond fetal life and contribute to long-term changes to physiology. It has been previously shown that the activity of certain one-carbon metabolising enzymes, including MTHFR, changes in response to both glucose and insulin metabolism (Dicker-Brown et al., 2001). Therefore, it is possible that any subtle changes in fetal glucose control throughout pregnancy could have resulted in this upregulation of *Mtrr* and *Mthfr* mRNA in the liver. We also demonstrate subtle decreases in *Mtrr* and *Mat2a* in the JZ of F2 fetuses of Restricted mothers. As mentioned previously, such changes may have contributed to deficits in maternal one-carbon metabolites. Alternatively, this may reflect compensatory adaptations driven by changes in maternal one-carbon metabolism. Indeed, these enzymes have previously been shown to be regulated by the availability of one-carbon metabolites (Chango et al., 2009; Christensen et al., 2015). To further understand the implications of changes in the expression of these enzymes, tissue concentrations of one-carbon metabolites would have been required. Unfortunately, protein concentrations or enzymatic activity of these enzymes were also not assessed. Without this additional data, we can only speculate as to the importance of these changes in mRNA expression. What we were able to assess was the expression of genes that are transcriptionally affected by changes in one-carbon metabolites. This included DNMTs and IGF2, both of which are transcriptionally regulated by the SAM:SAH ratio (Koukoura et al., 2012). We found that only *Dnmt3b* was decreased in the male JZ. Given that both DNMT3A and DNMT3B, are involved in *de novo*

DNA methylation (Smith et al., 2024), this subtle change most probably does not contribute to major deficits in methylation capacity. The consequences of such potential changes to DNA methylation were not assessed in the present study.

## Antihyperglycaemic medication and one-carbon metabolism

Metformin and insulin treatment did not impact maternal weight gain, body composition, or fetal and placental weights, therefore initially supporting the clinical evidence of their overall safety during pregnancy (Rowan et al., 2008). Despite these crude markers being unchanged, we wanted to investigate changes to processes that might be responsible for more subtle deficits, which could impact offspring once they reach adulthood. First, we investigated the effects of metformin and insulin on one-carbon metabolism in F1 Restricted dams and found that, despite exerting their antihyperglycaemic effect through very different mechanisms, both reduced maternal plasma concentrations of SAM. However, only insulin treatment increased homocysteine and SAH. Metformin also subtly increased the liver expression of *Achy* and decreased *Mthfr*. Both are important in the remethylation process and probably contributed to the reduced levels of methionine and, subsequently, SAM formation. Ultimately, both metformin and insulin treatment resulted in a decrease in the SAM/SAH ratio and therefore methylation capacity. These changes are in comparison to the Restricted group; thus, it appears that these antihyperglycaemic medications have worsened maternal one-carbon metabolism more so than being born growth restricted alone. We would hypothesise, however, that, although we observed an additive effect in the present study, this outcome is not limited to growth restricted animals and metformin would induce similar effects on one-carbon metabolism in normal rats.

There is a plethora of research investigating the impact of metformin on various components of the one-carbon cycle, but many of these studies were performed in cell lines using higher doses than are clinically relevant. Cuyàs et al. (2018) demonstrated that metformin promotes global methylation by decreasing SAH levels. This could be argued to be a result of the high metformin exposure used in this study (1 mmol $L^{-1}$ for 2 days). Previous work in *Caenorhabditis elegans* found similar results to ours, showing that metformin treatment decreases SAM at the same time as increasing SAH levels and therefore causes an overall reduction in the SAM/SAH ratio (Xiao et al., 2022). Furthermore, no studies to date have investigated the impact of metformin on one-carbon metabolism in pregnant animals

Fewer studies have investigated the impact of insulin on one-carbon metabolism. One study in hepatic cells treated with insulin found an increase in homocysteine remethylation, consequently increasing intracellular SAM concentrations by inducing methionine adenosyltransferase activity (Chiang et al., 2009), which conflicts with our findings of reduced SAM in maternal circulation. However, aligning with our findings, insulin increased cellular homocysteine production primarily by its inhibition of transsulfuration (Chiang et al., 2009). This inhibition of the transsulfuration pathway could explain the dramatic increase in maternal SAH concentrations as a result of insulin treatment.

As with the investigation of the impact of Restriction alone, we then explored how antihyperglycaemic medication impacted the expression of placental one-carbon metabolism enzymes. The reduction in *Mtr*, *Mthfr* and *Mat2a* in the JZ in response to metformin treatment may have contributed to reduced substrate availability and subsequent reduction in remethylation capacity in the maternal plasma, despite the lack of change in gene expression in the maternal liver (Tessari et al., 2005). Alternatively, changes to the expression of these enzymes may occur in response to changes to tissue glucose or insulin exposure and, if enzyme activity follows the change in mRNA expression, could impact local availability of substrates in the JZ. We also demonstrated that insulin treatment induced many effects similar to metformin, including decreased expression of *Mthfr*, *Mtr*, *Mtrr* and *Mat2a* in the JZ. These findings are again in agreement with Chiang et al. (2009), who found that the expression of these genes was decreased in insulin-exposed HepG2 cells: although how insulin mediated this is unknown. Dicker-Brown et al. (2001) also demonstrated that increasing insulin concentration led to a stepwise decrease in the MTHFR activity of HepG2 cells. It is possible that the mechanism by which metformin alters one-carbon metabolism is by increasing tissues insulin sensitivity, in which case its effects may be similar to that of insulin.

Antihyperglycaemic treatment also decreased the placental expression of *Dnmt1*, *Dnmt3a* and *Dnmt3b*. Although DNA methylation was not assessed, the reduction in expression of these genes alongside reduced *Igf2* expression, which is often used as a marker of placental methylation (Koukoura et al., 2012), in the presence of an altered SAM/SAH ratio suggests perturbed methylation of select genes. Previous studies have shown that a reduction in methyl group sources (i.e. folates) in pregnant rodents causes global DNA hypomethylation in the placenta as a result of reduced methylation capacity (Kim et al., 2009). Our findings, therefore, suggest that by downregulating one-carbon metabolism, antihyperglycaemic drugs may contribute to placental hypomethylation and changes in mRNA expression.

Similar to what was detected in Restricted *vs.* Control fetuses, fetuses exposed to antihyperglycaemic medication had no changes in circulating one-carbon metabolites. However, metformin and insulin impacted expression of key one-carbon regulating genes in the fetal liver and altered pancreatic $\alpha$-cell number, suggesting an indirect effect on fetal outcomes. Liver expression of *Mtr*, *Mat1a*, *Achy* and *Mthfr* was decreased by treatment with insulin exposure, impacting outcomes significantly in female fetuses. Treatment decreased *Dnmt1* and *Dnmt3b* expression with insulin reducing expression to a greater degree than metformin. This outcome is highly interesting given that insulin from the maternal system does not pass through the placenta (Ruiz-Palacios et al., 2017), whereas metformin is assumed to readily cross the placenta (Vanky et al., 2005). As such, the effect of antihyperglycaemic medication on fetal liver expression of one-carbon metabolising genes is more probably related to subtle changes in nutrients or placental regulators of fetal development. Despite neither metformin or insulin treatment impacting maternal glucose or insulin values during the GTT, it is possible that they were impacted by treatment immediately following dosing. It is important to note that the antihyperglycaemic medications were not provided on the morning of the GTTs because we had wanted to focus on understanding how these drugs impacted chronic outcomes rather than their acute effects. Pharmacokinetic studies have demonstrated that plasma concentrations of metformin decrease by 99% within 5 h of oral treatment (Choi et al., 2006), whereas the effects of s.c. insulin glargine injection on blood glucose levels are no longer different 8 h after exposure (Lu et al., 2020). Thus, treatment probably induced temporary changes in glucose or insulin concentrations, which were not reflected in the collected blood samples, and which may have impacted one-carbon metabolism as well as liver and pancreatic development.

### Growth restriction, antihyperglycaemic medication and F2 outcomes

A key finding of the present study was that Restriction induced subtle changes to $\beta$-cell number in the developing fetuses, which may contribute to the $\beta$-cell deficit that is known to occur in the F2 offspring once they reach adulthood (Cheong et al., 2016). Specifically, we have previously found that pancreatic $\beta$-cell mass at 6 months post-delivery was transiently reduced in F2 males of mothers born growth-restricted with no difference by 12 months, whereas pancreatic $\beta$-cell mass was not different in F2 females at 6 and 12 months (Cheong et al., 2016). In the present study, a significant increase in the F2 fetal $\beta$-cell/islet ratio was found at E20 in the Restricted group, suggesting either increased and/or accelerated

$\beta$-cell development, with no clear sex differences. This is consistent with the limited literature that suggests that mouse offspring exposed to the metabolic perturbations of diabetes in pregnancy show accelerated $\beta$-cell and $\alpha$-cell development at delivery (Zou et al., 2024). Interestingly, they found a subsequent slowdown in the development of $\beta$-cells and $\alpha$-cells during the postnatal period such that, by postnatal day 14, there were similar proportions of $\beta$-cells and $\alpha$-cells in the experimental and control groups (Zou et al., 2024). It is possible that, with further follow-up, reduced $\beta$-cell mass may eventually emerge.

In Restricted rats that were given antihyperglycaemic medication, we once again detected no changes in fetal metabolic control. Furthermore, it appears that metformin and insulin treatment does not restore the observed changes in $\beta$-cell/islet ratio. Instead, we reported an increase in $\alpha$-cell proportional area in female fetuses exposed to metformin. Metformin has been shown previously in one study to increase both $\alpha$-cell and total islet cell number compared to adult diabetic control rats (Xu et al., 2022) but the present study is the first to show that metformin treatment during pregnancy impacts fetal $\alpha$-cell area. Although neither insulin or glucagon levels in fetal circulation were affected by metformin in the present study, the long-term effect on offspring glucagon secretion should be investigated in future studies.

### Limitations

One key limitation of the present study was that the effects of metformin and insulin treatment on one-carbon metabolism and fetal outcomes were only assessed in Restricted animals, and these animals did not develop hyperglycaemia. Although it is of interest that these treatments exacerbated the effects of growth restriction on one-carbon metabolism, we anticipate that these drugs would induce similar effects in control animals. Furthermore, it is difficult to use data from the present study to inform on the risk of metformin or insulin treatment for fetal development and long-term outcomes when the control group was not diabetic. It is possible that the deficits caused by antihyperglycaemic medication are not as severe as the effects of diabetes itself and so data should be interpreted with caution. We also acknowledge the lack of data regarding protein concentrations of one-carbon metabolising enzymes or a measure of enzymatic activity as a key limitation of this study. Tissue concentrations of one-carbon metabolites also could not be assessed. Without these data, the conclusions surrounding the importance of the measured changes in mRNA expression and circulating one-carbon metabolite concentrations in both dams and fetuses are limited. Thus, future studies will investigate this to further elucidate the full impact of both Restriction and anti-hyperglycaemic medication on one-carbon metabolism in pregnancy and therefore the potential role this cycle plays in the transmission of metabolic disease.

## Conclusions

Overall, in dams born growth restricted, as well as in those treated with antihyperglycaemic medication, it appears that fetuses are largely protected from any major changes in maternal dysregulation of one-carbon metabolism. There also appears to be limited impacts on fetal metabolism as a result of these prenatal environments; however given the changes in gene expression of key enzymes in the fetal liver, potential alterations in the methylation state of the placenta and evidence of some changes in pancreatic development, it is possible these fetuses will still have a predisposition to cardiometabolic disease. The present study therefore highlights that intergenerational disease could be partially mediated by changes to one-carbon metabolism as a result of maternal growth restriction and that antihyperglycaemic medications may exacerbate this dysregulation. It is therefore important that further studies are conducted to explore the long-term impacts of these changes.

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

## Additional information

### Data availability statement

Data are available from the corresponding author upon reasonable request.

### Competing interests

The authors declare that they have no competing interests.

### Author contributions

J.F.B., K.M.M., S.C., M.E.W. and J.S.M.C. designed the study. J.F.B., S.G., D.A.Z., A.K.K.T., D.K., D.D., L.Y.L. and V.K. performed all experiments. All authors were involved in the analysis and interpretation of the data. All authors were involved in drafting the manuscript and revising it critically for intellectual content. All authors approve of the final version of the manuscript submitted for publication. All persons designated as authors qualify for authorship, and all those who qualify for authorship are listed.

### Funding

This research was supported by the University of Melbourne and the University of Queensland. This grant was also supported by the Allen foundation (2018.562). JFB held a Faculty of Medicine, Dentistry and Health Sciences Postdoctoral Fellowship at the University of Melbourne. KMM was funded by a NHMRC Senior Research Fellowship (APP1078164). DAZ was supported by an Australian Government Research Training Program (RTP) Scholarship.

### Acknowledgements

Open access publishing facilitated by The University of Queensland, as part of the Wiley - The University of Queensland agreement via the Council of Australian University Librarians.

### Author's present address

M. E. Wlodek: Department of Obstetrics, Gynaecology and Newborn Health, The University of Melbourne, Parkville, VIC, Australia.

### Keywords

B12, beta cell, GDM, methionine, SAM:SAH ratio

### Supporting information

Additional supporting information can be found online in the Supporting Information section at the end of the HTML view of the article. Supporting information files available:

**Peer Review History**

