## [Peer Review History · The Journal of Physiology]

Metformin and insulin exacerbate one-carbon metabolism deficits in pregnant growth restricted rats and impacts the placenta, fetal liver and pancreas

Dayna A Zimmerman, Jessica F Briffa, Dewei Kong, Sogand Gravina, Venea Dara Daygon, Vinod Kumar, Karen M Moritz, Lillian Y Lim, Adrian KK Teo, Shiao-Yng Chan, Mary E Wlodek, and James S M Cuffe

DOI: 10.1113/JP289453

Corresponding author(s): James Cuffe (j.cuffe1@uq.edu.au)

The following individual(s) involved in review of this submission have agreed to reveal their identity: Ellen Menkhorst (Referee #2)

Review Timeline:	Submission Date:	11-Jun-2025
	Editorial Decision:	08-Jul-2025
	Revision Received:	21-Jul-2025
	Accepted:	05-Aug-2025

Senior Editor: Kim Barrett

Reviewing Editor: Max Petersen

Transaction Report:

Dear Dr Cuffe,

Re: JP-RP-2025-289453 "**Metformin and insulin exacerbate one-carbon metabolism deficits in pregnant growth restricted rats and impacts the placenta, fetal liver and pancreas**" by Dayna A Zimmerman, Jessica F Briffa, Dewei Kong, Sogand Gravina, Dara Daygon, Vinod Kumar, Karen M Moritz, Lillian Y Lim, Adrian KK Teo, Shiao-Yng Chan, Mary E Wlodek, and James S M Cuffe

Thank you for submitting your manuscript to The Journal of Physiology. It has been assessed by a Reviewing Editor and by 2 expert referees and we are pleased to tell you that it is acceptable for publication following satisfactory revision.

REVISION CHECKLIST:

We look forward to receiving your revised submission.

Yours sincerely,

Kim Barrett
Senior Editor
The Journal of Physiology

REQUIRED ITEMS

- Author photo and profile. First or joint first authors are asked to provide a short biography (no more than 100 words for one author or 150 words in total for joint first authors) and a portrait photograph. These should be uploaded and clearly labelled together in a Word document with the revised version of the manuscript. See Information for Authors for further details.

- Papers must comply with the Statistics Policy: https://jp.msubmit.net/cgi-bin/main.plex?form_type=display_requirements#statistics.

In summary:

- If $n \leq 30$, all data points must be plotted in the figure in a way that reveals their range and distribution. A bar graph with data points overlaid, a box and whisker plot or a violin plot (preferably with data points included) are acceptable formats.

- If $n > 30$, then the entire raw dataset must be made available either as supporting information, or hosted on a not-for-profit repository, e.g. FigShare, with access details provided in the manuscript.

- 'n' clearly defined (e.g. x cells from y slices in z animals) in the Methods. Authors should be mindful of pseudoreplication.

- All relevant 'n' values must be clearly stated in the main text, figures and tables.

- The most appropriate summary statistic (e.g. mean or median and standard deviation) must be used. Standard Error of the Mean (SEM) alone is not permitted.

- Exact p values must be stated. Authors must not use 'greater than' or 'less than'. Exact p values must be stated to three significant figures even when 'no statistical significance' is claimed.

- Please include an Abstract Figure file, as well as the Figure Legend text within the main article file. The Abstract Figure is a piece of artwork designed to give readers an immediate understanding of the research and should summarise the main conclusions. If possible, the image should be easily 'readable' from left to right or top to bottom. It should show the physiological relevance of the manuscript so readers can assess the importance and content of its findings. Abstract Figures should not merely recapitulate other figures in the manuscript. Please try to keep the diagram as simple as possible and without superfluous information that may distract from the main conclusion(s). Abstract Figures must be provided by authors no later than the revised manuscript stage and should be uploaded as a separate file during online submission labelled as File Type 'Abstract Figure'. Please also ensure that you include the figure legend in the main article file. All Abstract Figures should be created using BioRender. Authors should use The Journal's premium BioRender account to export high-resolution images. Details on how to use and access the premium account are included as part of this email.

EDITOR COMMENTS

Reviewing Editor:

Methods Details:

See minor comments made by referee #1.

Thank you for submitting your work to JP. The referees have identified only minor specific issues to address in revision, but both noted that the text is too long, making it difficult to read and extract the key results and conclusions. Careful editing for concision is recommended.

REFEREE COMMENTS

Referee #1:

This is an interesting study examining the impact of common anti-glycemic drugs metformin and insulin on one-carbon metabolism during pregnancy. Overall, this study provides new information about the mechanism of action of these drugs during pregnancy and potentially identifies the importance of understanding the clinical history of a patient before prescribing drugs.

However, the study design makes it difficult to ascertain the impact of metformin/insulin on placenta/fetal one-carbon metabolism. In this study, it seems like in the end, fetal growth restriction was used as a tool to create hyperglycemic pregnant F1 dams (although in this particular cohort the restricted F1 dams weren't actually hyperglycemic). This means, that from this study, it is not possible to say whether these drugs specifically affect one-carbon metabolism in restricted dams or whether they would affect all dams in the same way. Are there other studies that do investigate the impact of metformin/insulin on one-carbon metabolism in dams that were not growth restricted?

This manuscript is very long and could benefit from editing for brevity. A lot of information is presented, particularly results where there was no change between treatment and control. Is it possible to move results that do not particularly add anything to the story into supplementary data? As it stands, it is difficult for the reader to tease out the important and interesting results.

The discussion likewise would benefit from editing for brevity.

Referee #2:

Overall, this is a very thorough animal study, performed on a fairly large scale while also working to reduce the number of animals utilised which is important. The methodologies are highly relevant and detailed well. The discussion is particularly well written, sufficiently detailing relevant findings and the potential implications for pregnant women, without overstating. The manuscript is technically sound but could benefit from a clearer and broader introduction and a more abbreviated results section. The main issue is this study has relied on mRNA expression, lacking data using protein or enzyme activity. However, considering the depth of the study and obvious limits in fetal tissues, this is appropriate in my opinion. Importantly, this limitation is pointed out by the authors. The take home message is that there is evidence to support intergenerational predisposition to cardiometabolic disorders from those born growth restricted, and that metformin could potentially exacerbate this when used during pregnancy. More work is clearly needed in this space, especially clinically.

This research is impactful, contributes to the understanding in this field using a highly relevant study design is and supported by a valid and meaningful conclusion.

Minor comments

Introduction:

- I found the introduction slightly confusing. It would benefit from a rewrite to improve the flow. Also, more consistency with terms used would help clarify.
- The introduction could also benefit from being trimmed down to more essential information.
- A further effort could be made to ensure the introduction and results are suitable for a more general audience as the journal requests.

Methods:

- The operative and post-operative monitoring of animals was not described as required by the journal.
- The origin of animals also wasn't detailed as required.
- Line 202: I assume these are F1 dams. Please say this. Also, how long were the dams in the Comprehensive Lab Animal Monitoring System?
- Under the section titled Physical activity, systolic blood pressure and metabolic profiling, please detail the methods in chronological order.
- Line 284: for calculations based on staining, how many slides per animal were used and how were these slides selected etc? This information is helpful.

Results:

- Table 1 & 2 would be better as supplementary tables.
- Considering there are many non-significant findings, I wonder if the results could be reduced in a way maintain highlighting significant values with more of a summary for the non-significant findings.

Discussion:

- Highly relevant and well written. Again, this section is quite lengthy considering the findings. Shortening this section would improve readability.

END OF COMMENTS

Dear Professor Barrett

We thank you and the reviewers for their valuable time reviewing this manuscript. We have made a number of changes to the paper based on this feedback and feel that the paper is now much improved. In particular, we have shortened the paper and rearranged the introduction to make it easier to follow. Please see below the response to individual comments.

Editors

1. Required items

We have checked the manuscript carefully for all required items and have provided an abstract figure with figure legend. We have also made sure to include details such that the n number is displayed in the figure legends as well as the text. We have also used the exact P value as often as possible in the manuscript. Importantly we have included a large amount of physiological data demonstrating that maternal physiology isn't changed in this model to highlight that physiological changes did not drive our phenotype. Most of this data is not significant, however we have included all exact P values in the tables.

Referee #1:

1. **The study design makes it difficult to ascertain the impact of metformin/insulin on placenta/fetal one-carbon metabolism. It is not possible to say whether these drugs specifically affect one-carbon metabolism in restricted dams or whether they would affect all dams in the same way.**

The current study was setup to investigate two related outcomes: the effects of growth restriction/GDM on one-carbon metabolism as well as how antihyperglycemic drugs impact one-carbon metabolism in Restricted animals. While we do show that metformin/insulin impacts one-carbon metabolism and fetal development, our study design means that this was only assessed in Restricted animals. We agree that this makes it difficult to ascertain the overall impact of these drugs in all pregnant animals as we did not specifically investigate this. Due to the ethical requirements for minimisation of animal numbers, we did not investigate the effects of these drugs in our sham animals. We acknowledge as a limitation, and we have now emphasised this in the limitations section of this paper. We do anticipate, that if we had investigated sham animals, that they would also have had decreases in one-carbon metabolism. We expect that these drugs would impact this pathway in all pregnant animals and may even impact this pathway in non-pregnant animals. We have also added comments to this in the discussion (line 643).

2. **Are there other studies that do investigate the impact of metformin/insulin on one-carbon metabolism in dams that were not growth restricted?**

We are the first study that have investigated the impact of metformin and insulin on one-carbon metabolism in pregnant rodents.

- 3. This manuscript is very long and could benefit from editing for brevity. A lot of information is presented, particularly results where there was no change between treatment and control. Is it possible to move results that do not particularly add anything to the story into supplementary data? As it stands, it is difficult for the reader to tease out the important and interesting results. The discussion likewise would benefit from editing for brevity.**

We agree that the paper as it was submitted was too long. We have edited it heavily to reduce the overall word count by over 1000 words. We have chosen to keep all of the data which demonstrated that growth restriction and antihyperglycemic medication had minimal impact on physiological pathways in the manuscript as this data is key to make sure that the phenotype we observe is not due to other factors. We have however, reworded the text to make these results more concise. If The Journal of Physiology permitted supplementary information, we would have included some of these as supplementary files. We have also shortened the discussion for conciseness.

Referee #2:

- 1. Overall, this is a very thorough animal study, performed on a fairly large scale while also working to reduce the number of animals utilised which is important. The methodologies are highly relevant and detailed well. The discussion is particularly well written, sufficiently detailing relevant findings and the potential implications for pregnant women, without overstating.**

We thank the reviewer for this overview and appreciation of our efforts for minimising the number animals required.

- 2. The manuscript is technically sound but could benefit from a clearer and broader introduction and a more abbreviated results section.**

We have now heavily edited the paper and rearranged the introduction to make is clearer to read. We have reduced the overall word count by more than 1000 words by streamlining the introduction, cutting some of the results section and reducing the length of the discussion.

- 3. The main issue is this study has relied on mRNA expression, lacking data using protein or enzyme activity. However, considering the depth of the study and obvious limits in fetal tissues, this is appropriate in my opinion. Importantly, this limitation is pointed out by the authors.**

We do acknowledge that this is one of the key limitations of this study.

- 4. The take home message is that there is evidence to support intergenerational predisposition to cardiometabolic disorders from those born growth restricted, and that metformin could potentially exacerbate this when used during pregnancy. More work is clearly needed in this space, especially clinically. This research is impactful, contributes to the understanding in this field using a highly relevant study design is and supported by a valid and meaningful conclusion.**

We thank the reviewer for this positive review.

Minor comments

- 5. I found the introduction slightly confusing. It would benefit from a rewrite to improve the flow. Also, more consistency with terms used would help clarify. The introduction could also benefit from being trimmed down to more essential information. A further effort could be made to ensure the introduction and results are suitable for a more general audience as the journal requests.**

We have performed a rewrite in order to improve flow for the reader. We have also gone through and standardised terms throughout the paper to improve clarity. This rewrite has also reduced the length of the introduction by approximately 150 words. Throughout the introduction and results we have also reworded key parts to make it more appropriate for a general audience.

- 6. The operative and post-operative monitoring of animals was not described as required by the journal.**

Outlined below are the complete details on operative and post-operative monitoring performed. Details have also been included within the manuscript.

Animals were monitored throughout surgery to ensure a proper depth of anaesthesia. Immediately following surgery, the animals are constantly observed until they have regained the righting reflex and were then monitored hourly on the day of surgery then twice daily for 3 days post-surgery. What we monitored specifically for was: appearance, behaviour, response to provocation/vocalisation, activity/movement, chromodacryorrhea (red discharge from eyes), respiration, blood in nest, and other sign(s). As we didn't want to disturb the rats further during delivery, this was only done by looking into the cage in the housing rack and we didn't weigh them to assess body weight changes.

- 7. The origin of animals also wasn't detailed as required.**

Outlined below are the complete details on the origin of the animals. Details have also been included within the manuscript (line 164).

Wistar-Kyoto rats were obtained from the Australian Resource Centre (Murdoch, WA, Australia).

- 8. Line 202: I assume these are F1 dams. Please say this. Also, how long were the dams in the Comprehensive Lab Animal Monitoring System?**

Yes, F1 dams. This has been updated in the manuscript (line 191).

- 9. Under the section titled Physical activity, systolic blood pressure and metabolic profiling, please detail the methods in chronological order.**

This has now been reordered within the manuscript.

- 10. Line 284: for calculations based on staining, how many slides per animal were used and how were these slides selected etc? This information is helpful.**

We have updated this in the methods section (lines 274-282)

11. Table 1 & 2 would be better as supplementary tables

We agree that these tables would be suitable as supplementary tables for journals that support supplementary tables. However, as The Journal of Physiology does not permit supplementary tables, we have left them in the main manuscript.

12. Considering there are many non-significant findings, I wonder if the results could be reduced in a way maintain highlighting significant values with more of a summary for the non-significant findings.

We have reduced the length of the results section significantly while still making sure we follow editorial rules for the journal. Key data demonstrating that Restriction or metformin had no impact on maternal physiology are still included (as it is actually important to highlight safety rather than just to focus on some negative outcomes). However, where we could, we have now grouped various parameters together and indicated no difference and referred to the relevant table.

13. Highly relevant and well written. Again, this section is quite lengthy considering the findings. Shortening this section would improve readability.

We do acknowledge that this section in particular was too long. We have significantly reduced the length of the discussion to improve readability.

Dear Dr Cuffe,

Re: JP-RP-2025-289453R1 "**Metformin and insulin exacerbate one-carbon metabolism deficits in pregnant growth restricted rats and impacts the placenta, fetal liver and pancreas**" by Dayna A Zimmerman, Jessica F Briffa, Dewei Kong, Sogand Gravina, Venea Dara Daygon, Vinod Kumar, Karen M Moritz, Lillian Y Lim, Adrian KK Teo, Shiao-Yng Chan, Mary E Wlodek, and James S M Cuffe

We are pleased to tell you that your paper has been accepted for publication in The Journal of Physiology.

Yours sincerely,

Kim Barrett
Senior Editor
The Journal of Physiology

If you would like to receive our 'Research Roundup', a monthly newsletter highlighting the cutting-edge research published in The Physiological Society's family of journals (The Journal of Physiology, Experimental Physiology, Physiological Reports, The Journal of Nutritional Physiology and The Journal of Precision Medicine: Health and Disease), please click this link, fill in your name and email address and select 'Research Roundup':
<https://www.physoc.org/journals-and-media/membernews>

- You can help your research get the attention it deserves! Check out Wiley's free Promotion Guide for best-practice recommendations for promoting your work at: www.wileyauthors.com/eo/guide. You can learn more about Wiley Editing Services which offers professional video, design, and writing services to create shareable video abstracts, infographics, conference posters, lay summaries, and research news stories for your research at: www.wileyauthors.com/eo/promotion.

EDITOR COMMENTS

Reviewing Editor:

Thank you for addressing the referee comments. Congratulations on this strong and impactful work.

REFEREE COMMENTS

Referee #1:

I am satisfied with this version and I appreciate the authors taking the time to

Referee #2:

Thank you for your considered comments to my review.